# Macrophages regulate gastrointestinal motility through complement component 1q

**Mihir Pendse[1], Haley De Selle[1], Nguyen Vo[1], Gabriella Quinn[1], Chaitanya Dende[1], Yun Li[1], Cristine N Salinas[1], Tarun Srinivasan[1], Daniel C Propheter[1], Alexander A Crofts[1], Eugene Koo[1], Brian Hassell[1], Kelly A Ruhn[1], Prithvi Raj[1], Yuuki Obata[1]\*, Lora V Hooper[1,2]\***

[1]Department of Immunology, The University of Texas Southwestern Medical Center, Dallas, United States; [2]The Howard Hughes Medical Institute, The University of Texas Southwestern Medical Center, Dallas, United States

**Abstract** Peristaltic movement of the intestine propels food down the length of the gastrointestinal tract to promote nutrient absorption. Interactions between intestinal macrophages and the enteric nervous system regulate gastrointestinal motility, yet we have an incomplete understanding of the molecular mediators of this crosstalk. Here, we identify complement component 1q (C1q) as a macrophage product that regulates gut motility. Macrophages were the predominant source of C1q in the mouse intestine and most extraintestinal tissues. Although C1q mediates the complement-mediated killing of bacteria in the bloodstream, we found that C1q was not essential for the immune defense of the intestine. Instead, C1q-expressing macrophages were located in the intestinal submucosal and myenteric plexuses where they were closely associated with enteric neurons and expressed surface markers characteristic of nerve-adjacent macrophages in other tissues. Mice with a macrophage-specific deletion of *C1qa* showed changes in enteric neuronal gene expression, increased neurogenic activity of peristalsis, and accelerated intestinal transit. Our findings identify C1q as a key regulator of gastrointestinal motility and provide enhanced insight into the crosstalk between macrophages and the enteric nervous system.

**\*For correspondence:**
yuki.obata@utsouthwestern.edu (YO);
lora.hooper@utsouthwestern.edu (LVH)

**Competing interest:** The authors declare that no competing interests exist.

## Editor's evaluation

This study provides a fundamental finding that complement C1q produced by enteric macrophages shapes neuronal function and gut motility. The authors present convincing data showing that while macrophage-derived C1q is not necessary for defenses against enteric pathogens, it plays an important role in regulating neuronal gene expression and intestinal transit. These findings will be of interest to gastroenterologists, neuroscientists and immunologists in revealing a novel neuroimmune axis in gut homeostasis.

## Introduction

Peristalsis is the physical force that propels food through the intestine, promoting digestion and nutrient absorption. The gastrointestinal motility that underlies peristalsis is a complex process that requires coordination of the activity of smooth muscle cells by enteric neurons (*Rao and Gershon, 2016*). Several studies have revealed that intestinal macrophages impact gastrointestinal motility by regulating the functions of enteric neurons and facilitating their interactions with smooth muscle cells (*Muller et al., 2014*; *Matheis et al., 2020*).

DOI: https://doi.org/10.7554/eLife.78558

Macrophages carry out diverse functions in the intestine that vary according to their anatomical location. For example, macrophages that localize to the tissue located directly underneath the gut epithelium — known as the lamina propria — contribute to immune defense against pathogenic bacteria (*Gabanyi et al., 2016*). A distinct group of macrophages localizes to the tissues located beneath the lamina propria, between the circular and longitudinal muscle layers in the tissue region known as the muscularis externa. These muscularis macrophages express genes that are distinct from lamina propria macrophages (*Gabanyi et al., 2016*). They directly regulate the activity of smooth muscle cells (*Luo et al., 2018*) and secrete soluble factors, such as bone morphogenetic protein 2 (BMP2), which interact with the enteric neurons that control smooth muscle activity (*Muller et al., 2014*). Muscularis macrophages thus play a key role in regulating gut motility. However, we have a limited understanding of the molecular mechanisms by which these macrophages regulate intestinal neuromuscular activity and gut motility.

C1q is a member of the defense collagen family that has distinct roles in immune defense and nervous system development and function (*Bossi et al., 2014*; *Casals et al., 2019*; *Shah et al., 2015*; *Thielens et al., 2017*). It is composed of six molecules each of C1qA, C1qB, and C1qC, forming a 410 kDa oligomer. C1q circulates in the bloodstream, where it participates in immune defense against infection by recognizing antibodies bound to invading bacteria. This binding interaction initiates the classical complement pathway, which entails the recruitment and proteolytic processing of other complement components that rupture the bacterial membrane and recruit phagocytic cells (*Kishore and Reid, 2000*; *Noris and Remuzzi, 2013*). C1q is also produced by microglia (brain-resident macrophage-like cells) in the brain where it promotes the pruning of neuronal synapses through an unclear mechanism (*Hammond et al., 2020*; *Hong et al., 2016*). Consequently, C1q deficiency results in heightened synaptic connectivity in the central nervous system which can lead to epilepsy (*Chu et al., 2010*).

C1q is also produced at barrier sites, such as the intestine, where encounters with commensal and pathogenic microbes are frequent. However, little is known about the physiological role of C1q in barrier tissues. Liver immune cells, including macrophages and dendritic cells, produce serum C1q; however, the cellular source of C1q in barrier tissues including the intestine remains unclear (*Petry et al., 2001*). Here, we show that C1q is produced by macrophages of the mouse intestine. Intestinal C1q-expressing macrophages exhibit properties of neuromodulatory macrophages from other tissues and are located close to enteric neurons that have a known role in controlling gut motility. Accordingly, mice lacking macrophage C1q exhibit altered expression of enteric neuronal genes, increased neurogenic peristaltic contractions, and accelerated gastrointestinal motility. These findings identify C1q as a key mediator of a neuroimmune interaction that regulates gut motility.

## Results

### C1q is expressed by macrophages in the mouse small intestine

Soluble defense collagens are an ancient, evolutionarily conserved family of antimicrobial proteins with shared structural features including a *C*-terminal globular head and a collagen-like region (*Casals et al., 2019*). Little is known about the function of defense collagens at mucosal barrier sites, where microbial encounter is frequent. Our initial goal in this study was to identify soluble defense collagens that are expressed by the mouse intestine and to assess their role in host defense. Therefore, we measured the expression of 18 defense collagen genes in the mouse small intestine and colon by RNA sequencing (RNA-seq). The most abundant soluble defense collagen transcripts in the small intestine and colon were those encoding C1qA, C1qB, and C1qC (*Figure 1A*; *Figure 1—figure supplement 1*).

Serum C1q is produced by liver dendritic cells, monocytes, and macrophages (*El-Shamy et al., 2018*). However, the cellular source(s) of C1q in peripheral tissues, including the intestine, is unknown. Quantitative PCR (qPCR) analysis of fluorescence-activated cell sorting (FACS)-sorted cell suspensions recovered from the small intestines of wild-type C57BL/6 mice revealed that *C1qa*, *C1qb*, and *C1qc* transcripts were most abundant in CD45$^+$ cells, which include all immune cells, as compared to CD45$^-$ cells, which encompass epithelial cells and other non-immune cells (*Figure 1B*). Furthermore, C1q transcripts and protein were most abundant in CD45$^+$ cells recovered from the subepithelial compartment, which includes both the lamina propria and muscularis, as compared to CD45$^+$ cells recovered from the intraepithelial compartment of the small intestine (*Figure 1C and D*). Thus, C1q is expressed

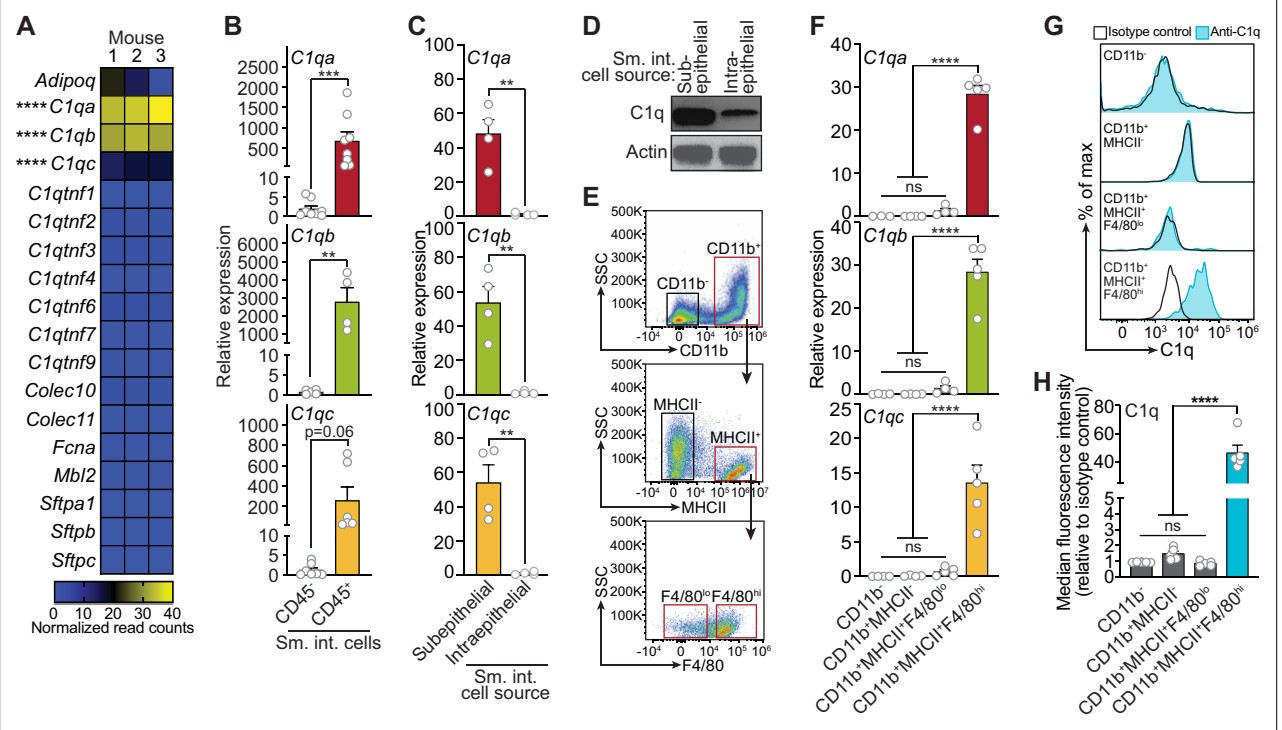

**Figure 1.** Complement component 1q (C1q) is expressed by macrophages in the mouse small intestine. (**A**) RNA-seq analysis of soluble defense collagen expression in the small intestines (ileum) of C57BL/6 mice. Data were adapted from a previously published RNA-seq analysis (*Gattu et al., 2019*). Data are available in the Gene Expression Omnibus repository under accession number GSE122471. Each column represents one mouse. (**B**) Quantitative PCR (qPCR) measurement of *C1qa*, *C1qb*, and *C1qc* transcript abundance in CD45[+] and CD45[-] cells purified from mouse small intestines by flow cytometry. Each data point represents one mouse, and the results are representative of two independent experiments. (**C**) qPCR measurement of *C1qa*, *C1qb*, and *C1qc* transcript abundance in subepithelial and intraepithelial cells recovered from mouse small intestines. Each data point represents one mouse, and the results are representative of three independent experiments. (**D**) Representative immunoblot of subepithelial and intraepithelial cells recovered from mouse small intestines, with detection of C1q and actin (control). Each lane represents cells from one mouse and the immunoblot is representative of three independent experiments. (**E**) Flow cytometry gating strategy for analysis of mouse small intestinal cell suspensions in panels F, G, and H. Cells were pre-gated as live CD45[+] cells. SSC, side-scatter; MHCII, major histocompatibility complex II. (**F**) qPCR measurement of *C1qa*, *C1qb*, and *C1qc* transcript abundance in cells isolated by flow cytometry from mouse small intestines as indicated in (**E**). Each data point represents cells pooled from three mice, and the results are representative of three independent experiments. (**G**) Flow cytometry analysis of intracellular C1q in small intestinal subepithelial cells identified as indicated in (**E**). (**H**) Quantitation of flow cytometry analysis in (**G**). Each data point represents one mouse, and the results are representative of two independent experiments. Sm. int., mouse small intestine; Error bars represent SEM. **p<0.01; ***p<0.001; ****p<0.0001; ns, not significant by one-way ANOVA (**A,F**) or two-tailed Student's *t*-test (**B,C,H**).

The online version of this article includes the following source data and figure supplement(s) for figure 1:

**Source data 1.** Unedited, uncropped immunoblot for *Figure 1D*.

**Figure supplement 1.** Complement component 1q (C1q) is expressed in the mouse colon.

by immune cells located in the subepithelial compartment of the intestine and is largely absent from epithelial cells and intraepithelial immune cells.

To identify intestinal immune cells that express C1q, we further analyzed the subepithelial CD45[+] cell population by flow cytometry. Expression of C1q transcripts and protein was highest in CD11b[+]MHCII[+]F4/80[hi] macrophages and was mostly absent from non-macrophage immune cells (*Figure 1E–H*). Thus, C1q is expressed by macrophages in the mouse small intestine.

## Macrophages are the primary source of C1q in the mouse gastrointestinal tract

We next assessed whether macrophages are the primary source of C1q in the intestine by analyzing two mouse models. First, we depleted macrophages by injecting neutralizing antibodies directed against the receptor for colony-stimulating factor 1 (CSF1R)(*Figure 2A*), which is required for the development of a subset of lamina propria macrophages (*Bogunovic et al., 2009*) and all muscularis

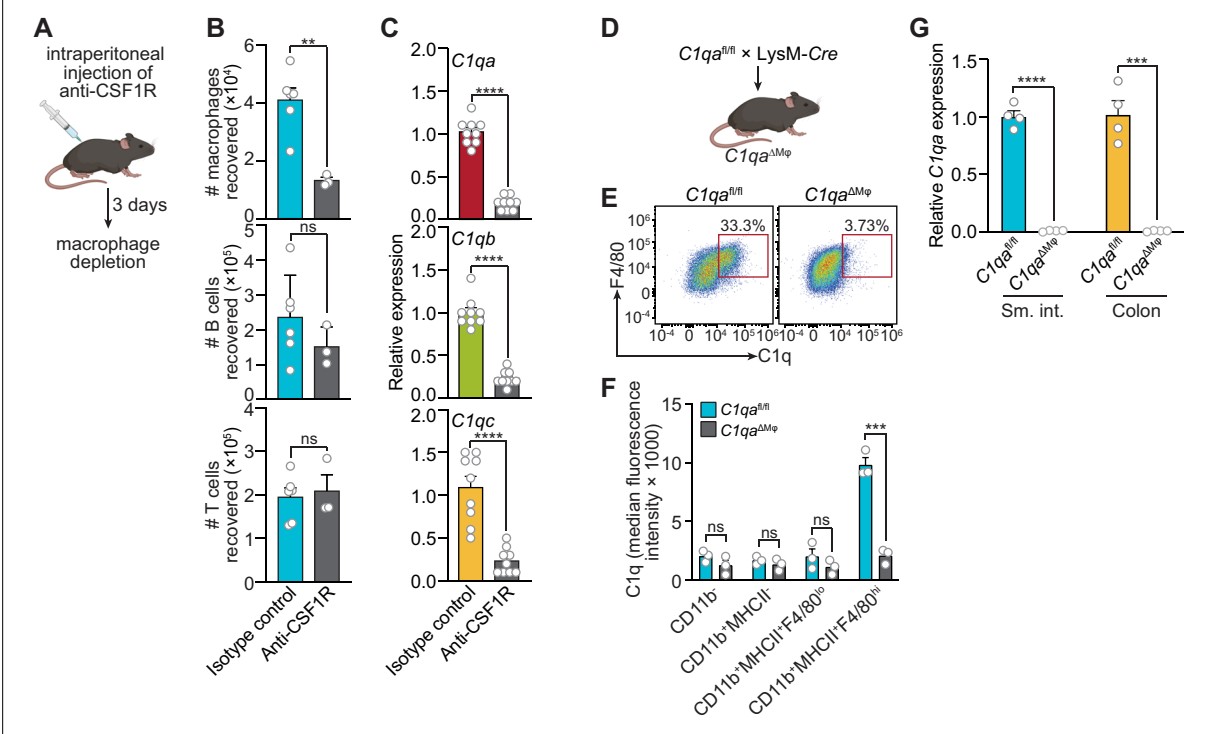

**Figure 2.** Macrophages are the primary source of complement component 1q (C1q) in the mouse gastrointestinal tract. (**A**) Macrophages were selectively depleted in C57BL/6 mice by intraperitoneal injection of anti-CSF1R antibody. Control mice were injected with isotype-matched non-specific antibodies. Mice were analyzed 72 hr after antibody injection. Panel was generated at Biorender.com. (**B**) Representative flow cytometry analysis of mouse small intestines after intraperitoneal injection of anti-CSF1R or isotype control antibody. All cells were gated as live CD45+. Macrophages were MHCII+ F4/80hi; B cells were CD19+; T cells were CD3+. Total small intestinal cell yields were $1.5 \times 10^6 \pm 4.9 \times 10^5$ cells. (**C**) Quantitative PCR (qPCR) measurement of *C1qa*, *C1qb*, and *C1qc* transcript abundance in mouse small intestines after intraperitoneal injection of anti-CSF1R or rat IgG2a (isotype control). Each data point represents one mouse and results are pooled from two independent experiments. (**D**) *C1qa*fl/fl mice were crossed with LysM-Cre transgenic mice to generate mice having a macrophage-selective deletion of *C1qa* (*C1qa*ΔMφ mice). Panel was generated at Biorender.com. (**E**) Representative flow cytometry analysis of intracellular C1q expression in small intestinal macrophages from *C1qa*fl/fl and *C1qa*ΔMφ mice. Mice were littermates from heterozygous crosses that remained co-housed. Cells were gated on live CD45+CD11b+MHCII+. (**F**) Quantitation of the flow cytometry analysis in (**E**). Each data point represents one mouse. Results are representative of two independent experiments. (**G**) qPCR measurement of *C1qa* transcript abundance in the small intestines (sm. int.) and colons of *C1qa*fl/fl and *C1qa*ΔMφ littermates. Each data point represents one mouse. Error bars represent SEM. **p<0.01; ***p<0.001; ****p<0.0001; ns, not significant by the two-tailed Student's *t*-test.

The online version of this article includes the following figure supplement(s) for figure 2:

**Figure supplement 1.** Complement component 1q (C1q) expression is lost systemically but preserved in the central nervous system of *C1qa*ΔMφ mice.

---

macrophages (***Muller et al., 2014***). Antibody injection led to a >twofold reduction in the number of macrophages recovered from the small intestine (***Figure 2B***), and a corresponding reduction in small intestinal C1q gene expression (***Figure 2C***), suggesting that macrophages are the primary source of intestinal C1q.

Second, we constructed a genetic model of C1q deficiency by crossing *C1qa*fl/fl mice (***Fonseca et al., 2017***) to mice carrying the *Lyz2-Cre* transgene (LysM-Cre mice), which is selectively expressed in myeloid cells including macrophages (***Figure 2D***). These mice, hereafter designated as *C1qa*ΔMφ mice, lacked C1q expression in intestinal macrophages (***Figure 2E and F***). Importantly, *C1qa*ΔMφ mice had markedly lower C1q expression in both the small intestine and colon (***Figure 2G***), indicating that macrophages are the main source of C1q in the intestine. Unexpectedly, the *C1qa*ΔMφ mice also lost *C1q* gene expression in the lung, skin, kidney, and liver (but not the brain), and the C1q protein was undetectable in the serum (***Figure 2—figure supplement 1***). These findings indicate that macrophages are the primary source of C1q in the intestine and suggest that LysM+ macrophages or macrophage-like cells are also the main sources of C1q in most extraintestinal tissues and the bloodstream.

## *C1qa*$^{\Delta M\varphi}$ mice do not show altered microbiota composition, barrier function, or resistance to enteric infection

The classical complement pathway is a well-studied host defense system that protects against systemic pathogenic infection (*Warren et al., 2002*; *Noris and Remuzzi, 2013*). Circulating C1q activates the complement pathway by binding to antibody-antigen complexes or to bacterial cell surface molecules, and thus protects against systemic infection. Therefore, we assessed whether C1q promotes the immune defense of the intestine.

We first determined whether C1q exhibits characteristics of known intestinal antimicrobial proteins, including induction by the intestinal microbiota and secretion into the gut lumen. *C1qa* was expressed at similar levels in the small intestines of germ-free and conventionally-raised mice (*Figure 3A*), suggesting that *C1q* expression is not induced by the gut microbiota. This contrasted with *Reg3g*, encoding the antimicrobial protein REG3G (*Cash et al., 2006*), which was expressed at a > twofold higher level in conventional as compared to germ-free mice (*Figure 3A*). Additionally, in contrast to REG3G, C1q was not detected in the gut lumen of either conventional or germ-free mice (*Figure 3B*). *C1qa* expression was also not markedly altered by a 24 hr oral infection with the intestinal pathogenic bacterial species *Salmonella* Typhimurium (*Figure 3C*). Although we cannot rule out the induction of C1q by longer-term pathogenic infections, these data indicate that C1q is not induced by the gut microbiota or by a 24 hr infection with *S. typhimurium*, in contrast to other intestinal antibacterial proteins.

We next assessed whether C1q regulates the composition of the gut microbiota. 16 *S* rRNA gene sequencing analysis of the fecal microbiotas of *C1qa*$^{fl/fl}$ and *C1qa*$^{\Delta M\varphi}$ mice showed that the microbiota composition was not appreciably altered in the absence of macrophage C1q (*Figure 3D*). Analysis of 16 *S* rRNA gene copy number in mesenteric lymph nodes further indicated no statistically significant differences in translocation of the microbiota to the mesenteric lymph nodes (*Figure 3E*). We next challenged *C1qa*$^{fl/fl}$ and *C1qa*$^{\Delta M\varphi}$ mice with dextran sulfate sodium (DSS), which damages the colonic epithelium and exposes underlying tissues to the commensal microbiota. However, the sensitivity of the *C1qa*$^{\Delta M\varphi}$ mice to DSS was similar to that of their *C1qa*$^{fl/fl}$ littermates as assessed by change in body weight and histopathological analysis (*Figure 3F*; *Figure 3—figure supplement 1*). There was also no change in intestinal paracellular permeability in *C1qa*$^{\Delta M\varphi}$ mice as measured by oral administration of FITC-dextran (*Figure 3G*). These results suggest that macrophage C1q does not substantially impact gut microbiota composition or intestinal epithelial barrier function.

To determine whether C1q protects against enteric infection we conducted oral infection experiments with the enteric pathogen *Citrobacter rodentium*. We chose *C. rodentium* as our model organism for two reasons. First, *C. rodentium* is a non-disseminating pathogen, allowing us to test specifically for C1q's role in intestinal infection. Second, *C. rodentium* clearance depends on immunoglobulins and complement component C3 (*Belzer et al., 2011*). Because C1q is bactericidal in concert with C3 and immunoglobulins, we predicted that *C1qa*$^{\Delta M\varphi}$ mice would be more susceptible to *C. rodentium* infection. However, *C1qa*$^{\Delta M\varphi}$ mice cleared *C. rodentium* similarly to their *C1qa*$^{fl/fl}$ littermates (*Figure 3H*) and showed similar histopathology (*Figure 3—figure supplement 2*), indicating that C1q is dispensable for defense against *C. rodentium* infection.

We also did not observe altered immunity in the absence of C1q. Measurement of transcripts encoding secreted immune effectors in the small intestines of *C1qa*$^{fl/fl}$ and *C1qa*$^{\Delta M\varphi}$ littermates revealed no statistically significant differences in cytokine expression (*Figure 3I*). Furthermore, there were no statistically significant differences in the percentages or absolute numbers of various T cell subsets, including $T_{helper}1$ ($T_H1$), $T_H2$, $T_H17$, and regulatory T ($T_{reg}$) cells between *C1qa*$^{fl/fl}$ and *C1qa*$^{\Delta M\varphi}$ mice (*Figure 3J*; *Figure 3—figure supplement 3*). Although total B cell numbers trended lower in *C1qa*$^{\Delta M\varphi}$ mice, the difference was not statistically significant (*Figure 3J*; *Figure 3—figure supplement 4*). There were also no statistically significant differences in the percentages or absolute numbers of total plasma cells (*Figure 3J*; *Figure 3—figure supplement 4*), IgA$^+$ plasma cells (*Figure 3J*; *Figure 3—figure supplement 4*), myeloid cells (*Figure 3J*; *Figure 3—figure supplement 5*), or innate lymphoid cells (*Figure 3J*; *Figure 3—figure supplement 6*) when comparing *C1qa*$^{fl/fl}$ and *C1qa*$^{\Delta M\varphi}$ mice. These results suggest that the absence of macrophage C1q has little impact on intestinal immunity. Altogether, our findings suggest that C1q does not participate substantially in intestinal immune defense and thus might have an intestinal function that is independent of its canonical role in activating the classical complement pathway.

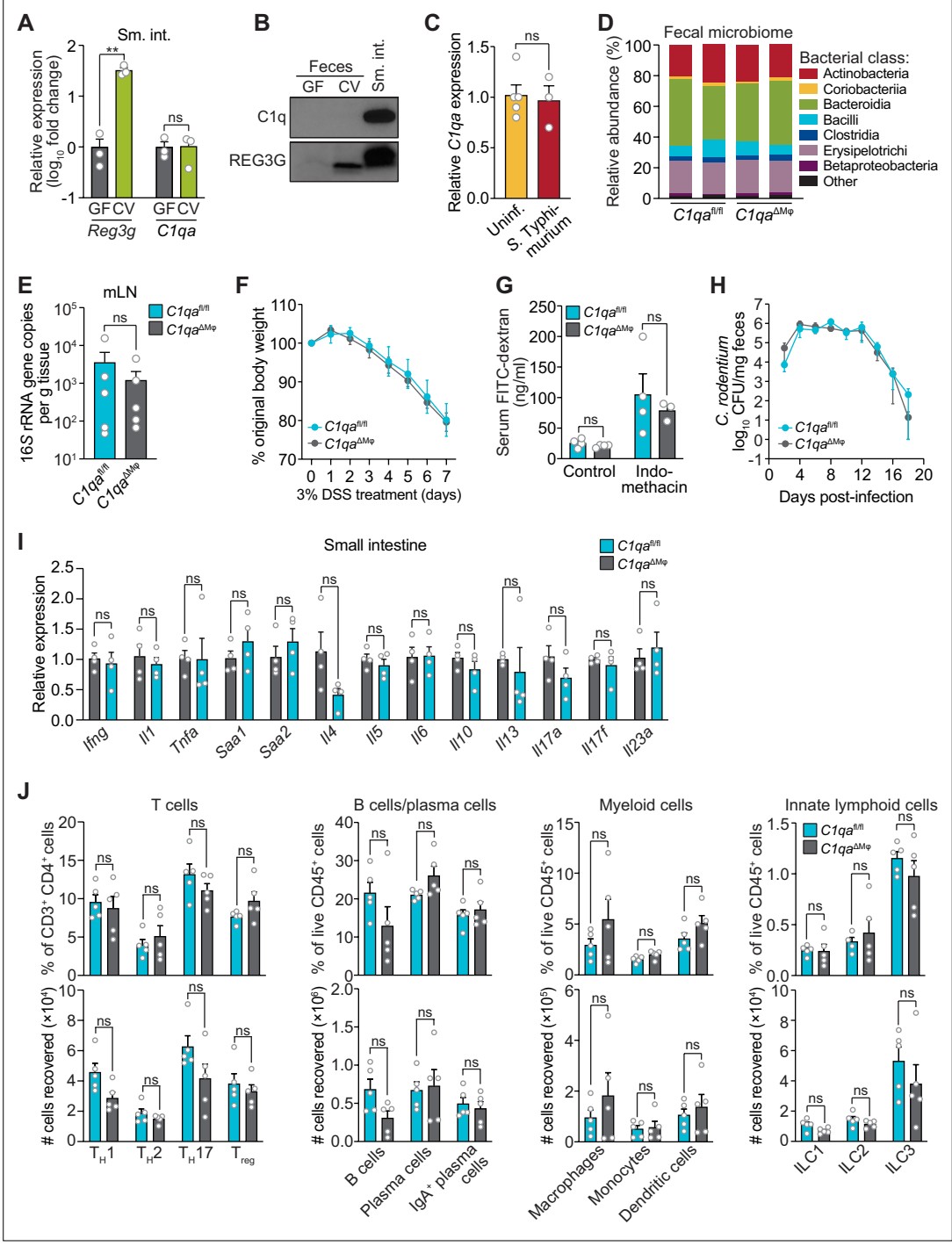

**Figure 3.** $C1qa^{\Delta M\varphi}$ mice do not show altered microbiota composition, barrier function, or resistance to enteric infection. (**A**) Small intestinal *C1qa* expression is not induced by the intestinal microbiota. Quantitative PCR (qPCR) measurement of *Reg3g* and *C1qa* transcript abundances in the small intestines of germ-free (GF) and conventional (CV) C57BL/6 mice. Each data point represents one mouse and the results are representative of two independent experiments. (**B**) C1q is not detected in the mouse intestinal lumen or feces. Representative immunoblot of an ammonium sulfate precipitation of intestinal luminal contents and feces from germ-free and conventional mice with detection of C1q. C1q in small intestinal tissue is shown for comparison at right. REG3G was analyzed as a control, as it is secreted into the intestinal lumen of conventional mice (*Cash et al., 2006*). Each lane represents multiple mice pooled (n=5 and 9 for germ-free and conventional, respectively) and the immunoblot is representative of three independent experiments. (**C**) C1q gene expression is not altered by acute enteric infection

*Figure 3 continued on next page*

*Figure 3 continued*

with *Salmonella typhimurium*. qPCR measurement of *C1qa* transcript abundance in small intestinal tissue after oral inoculation of mice with $10^9$ colony-forming units of *S. typhimurium* strain SL1344. Each data point represents one mouse, and the results are representative of two independent experiments. (**D**) Intestinal microbiota composition is not altered in *C1qa*$^{ΔMφ}$ mice. Phylogenetic analysis of 16 *S* rRNA gene sequences from fecal pellets collected from *C1qa*$^{fl/fl}$ and *C1qa*$^{ΔMφ}$ littermates. Operational taxonomic units with an average of 100 reads and populations greater than or equal to 1% were included in the graphical analysis. Each bar represents one mouse. Data are available from the Sequence Read Archive under BioProject ID PRJNA793870. (**E**) *C1qa*$^{ΔMφ}$ mice do not show altered translocation of bacteria to mesenteric lymph nodes (mLN). 16 *S* rRNA gene copy numbers were measured by qPCR with reference to a standard curve. Each data point represents one mouse. (**F**) *C1qa*$^{ΔMφ}$ mice do not show altered susceptibility to dextran sulfate sodium (DSS)-induced colitis. Mice were provided with 3% DSS in drinking water and body weights were monitored for 7 days. n=4 and 6 for *C1qa*$^{fl/fl}$ and *C1qa*$^{ΔMφ}$ littermates, respectively. Differences at each time point were not significant by the two-tailed Student's *t*-test. (**G**) *C1qa*$^{ΔMφ}$ mice do not show altered intestinal permeability. To measure intestinal permeability, *C1qa*$^{fl/fl}$ and *C1qa*$^{ΔMφ}$ littermates were gavaged with fluorescein isothiocyanate (FITC)-dextran (4 kDa), and serum FITC-dextran levels were determined by fluorescence microplate assay against a FITC-dextran standard curve. Indomethacin induces intestinal damage in mice and was used as a positive control. Each data point represents one mouse. (**H**) Time course of fecal *Citrobacter rodentium* burden following oral gavage of *C1qa*$^{fl/fl}$ and *C1qa*$^{ΔMφ}$ mice with 5×$10^8$ colony forming units (CFU) of *C. rodentium*. n=5 and 5 for *C1qa*$^{fl/fl}$ and $^{C1qaΔMφ}$ littermates, respectively. Differences at each time point were not significant by the two-tailed Student's *t*-test. (**I**) qPCR measurement of transcripts encoding secreted immune effectors in the small intestines of *C1qa*$^{fl/fl}$ and *C1qa*$^{ΔMφ}$ littermates. Each data point represents one mouse. (**J**) Flow cytometry analysis of small intestinal immune cell subsets from *C1qa*$^{fl/fl}$ and *C1qa*$^{ΔMφ}$ littermates. Gating strategies are shown in *Figure 3—figure supplement 1* through 4. ILC, innate lymphoid cell. Total small intestinal cell yields were $8.8 × 10^6 ± 2.9 × 10^6$ cells. Each data point represents one mouse. Sm. int., small intestine. Error bars represent SEM. **p<0.01; ns, not significant by the two-tailed Student's *t*-test.

The online version of this article includes the following source data and figure supplement(s) for figure 3:

**Source data 1.** Unedited, uncropped immunoblot for *Figure 3B*.

**Figure supplement 1.** Histological analysis of dextran sulfate sodium (DSS)-treated mice.

**Figure supplement 2.** Colon histology of *Citrobacter rodentium*-infected mice.

**Figure supplement 3.** Flow cytometry gating strategy for comparison of T cell populations in *C1qa*$^{fl/fl}$ and *C1qa*$^{ΔMφ}$ mice.

**Figure supplement 4.** Flow cytometry gating strategy for comparison of B cell and plasma cell populations in *C1qa*$^{fl/fl}$ and *C1qa*$^{ΔMφ}$ mice.

**Figure supplement 5.** Flow cytometry gating strategy for comparison of myeloid cell populations in *C1qa*$^{fl/fl}$ and *C1qa*$^{ΔMφ}$ mice.

**Figure supplement 6.** Flow cytometry gating strategy for comparison of innate lymphoid cell populations in *C1qa*$^{fl/fl}$ and *C1qa*$^{ΔMφ}$ mice.

## C1q is expressed by muscularis macrophages that are located near enteric neurons

Intestinal macrophages perform distinct functions depending on their anatomical location. Macrophages in the lamina propria protect against invasion by pathogenic microbes and promote tissue repair (*Grainger et al., 2017*). In contrast, muscularis macrophages that reside in deeper intestinal tissues, such as the muscularis externa (*Figure 4A*), regulate enteric neurons and smooth muscle cells that drive gastrointestinal motility (*De Schepper et al., 2018a*; *De Schepper et al., 2018b*). Furthermore, C1q has several well-described functions in regulating the development and activity of neurons of the central nervous system (*Hammond et al., 2020*; *Hong et al., 2016*), suggesting that intestinal C1q$^+$ macrophages might interact with enteric neurons. These prior findings prompted us to characterize the anatomical localization of C1q$^+$ macrophages within mouse intestinal tissues.

The enteric nervous system is a network of neurons whose cell bodies are organized into two regions of the gastrointestinal tract: the submucosal plexus and the myenteric plexus (*Figure 4A*). Immunofluorescence microscopy revealed that C1q was localized close to submucosal plexus nerve fibers marked with βIII tubulins (TUBB3) in *C1qa*$^{fl/fl}$ mice (*Figure 4B and C*). C1q was absent in *C1qa*$^{ΔMφ}$ mice despite the presence of similar overall numbers of CD169$^+$ macrophages (*Figure 4—figure supplement 1A*). Although C1q immunoreactivity in the myenteric plexus was less pronounced, flow

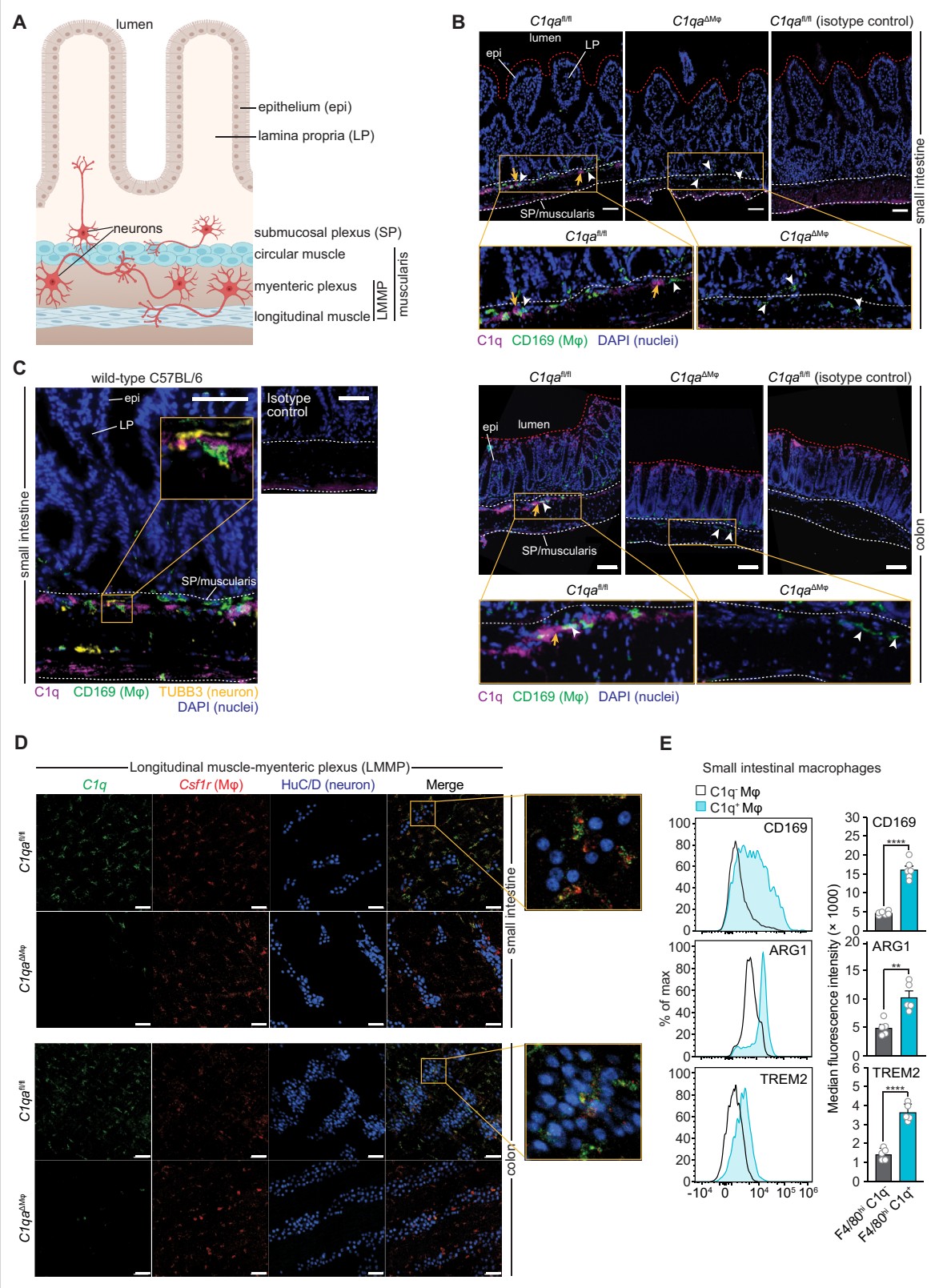

**Figure 4.** Complement component 1q (C1q) is expressed by muscularis macrophages that are located near enteric neurons. (**A**) Graphic depicting the muscularis of the mouse small intestine. The lumen, epithelium (epi), lamina propria (LP), submucosal plexus (SP), and longitudinal muscle-myenteric plexus (LMMP) are indicated. Created at Biorender.com. (**B**) Immunofluorescence detection of C1q (violet) and macrophages marked with CD169 (green) in the small intestine and colon of *C1qa*fl/fl and *C1qa*ΔMφ littermates. Nuclei were detected with 4',6-diamidino-2-phenylindole (DAPI; blue). Detection

*Figure 4 continued on next page*

*Figure 4 continued*

with isotype control antibodies on *C1qa*^fl/fl small intestines is shown at right. Anti-rat IgG AlexaFluor 488 and streptavidin-Cy5 were used as secondary stains for CD169 and C1q, respectively. The intestinal surface is denoted with a red dotted line and the gut lumen, epithelium, and lamina propria are indicated. The approximate region encompassing the submucosal plexus and the muscularis is denoted with two white dotted lines. Examples of C1q^+ areas are indicated with yellow arrows and examples of CD169^+ macrophages are indicated with white arrowheads. Note that the violet staining near the bottom of the muscularis is non-specific, as indicated by its presence in the isotype control image. Images are representative of three independent experiments. Scale bars = 50 µm. (**C**) Immunofluorescence detection of C1q (violet), macrophages marked with CD169 (green), and neurons marked with TUBB3 (yellow) in the small intestines of wild-type C57BL/6 mice. Nuclei are detected with DAPI (blue). The epithelium and lamina propria are indicated. The approximate region encompassing the submucosal plexus and the muscularis is denoted with two white dotted lines. The expanded image area delineated by a yellow square shows an example of the close association between C1q and TUBB3^+ neurons. Images are representative of images captured from three mice. Anti-rat IgG AlexaFluor 488, anti-rabbit IgG AlexaFluor 594, and streptavidin-Cy5 were used as secondary stains for CD169, TUBB3, and C1q, respectively, and an isotype control image is shown at upper right. Scale bars = 50 µm. (**D**) RNAscope detection of *C1qa* (green), muscularis macrophages marked by *Csf1r* (red), and immunofluorescence detection of enteric neuronal ganglia by HuC/D (blue) in LMMP wholemounts of small intestines and colons from *C1qa*^fl/fl and *C1qa*^ΔMφ mice. The expanded area denoted by a yellow square shows a close association between *C1qa*-expresssing muscularis macrophages and enteric neurons. Images are representative of three independent experiments. Scale bars = 50 µm. (**E**) C1q^+ intestinal macrophages express genes that are characteristic of nerve-adjacent macrophages. Flow cytometry analysis of CD169, Arginase 1, and TREM2 on C1q^- and C1q^+ macrophages recovered from the small intestines of wild-type C57BL/6 mice. Median fluorescence intensities from multiple mice are quantified in the panels at the right. Each data point represents one mouse (n=5–6 mice), and the results are representative of two independent experiments. Error bars represent SEM. **p<0.01; ****p<0.0001 by the two-tailed Student's *t*-test. Epi, epithelium; LP, lamina propria; SP, submucosal plexus; Mφ, macrophage; DAPI, 4',6-diamidino-2-phenylindole, LMMP, longitudinal muscle-myenteric plexus. Error bars represent SEM. ns, not significant by the two-tailed Student's *t*-test.

The online version of this article includes the following figure supplement(s) for figure 4:

**Figure supplement 1.** Flow cytometry analysis of complement component 1q (C1q) and CD169 expression on small intestinal macrophages.

cytometry analysis indicated that C1q was expressed by macrophages recovered from the muscularis (*Figure 4—figure supplement 1B*), which encompasses the myenteric plexus. Supporting this finding, RNAscope analysis of longitudinal muscle-myenteric plexus (LMMP) wholemounts revealed *C1qa*-expressing macrophages next to HuC/D^+ neurons (*Figure 4D*). Consistent with other validation data (*Figure 2E–G*), *C1qa* signals were mostly absent in muscularis macrophages of *C1qa*^ΔMφ mice. Finally, C1q-expressing intestinal macrophages showed elevated expression of Arginase 1, CD169, and TREM2 (triggering receptor expressed on myeloid cells 2) (*Figure 4E*), which are enriched on macrophages with known neuromodulatory functions (*Colonna, 2003*; *Paloneva et al., 2002*; *Ural et al., 2020*). Thus, C1q-expressing intestinal macrophages are located near enteric neurons in the submucosal and myenteric plexuses and express proteins that are characteristic of nerve-adjacent macrophages in other tissues.

## Numbers of enteric neurons are similar in *C1qa*^fl/fl and *C1qa*^ΔMφ mice

Gut macrophages engage in crosstalk with the enteric nervous system and regulate functions, including gastrointestinal motility, that depend on the enteric nervous system (*Muller et al., 2014*). This crosstalk involves the exchange of specific proteins such as bone morphogenetic protein 2 (BMP2) (*Muller et al., 2014*). Furthermore, microglial C1q promotes central nervous system development while also regulating neuronal transcriptional programs (*Benavente et al., 2020*; *Schafer et al., 2012*; *Stevens et al., 2007*). Given that intestinal C1q^+ macrophages phenotypically resemble peripheral neuromodulatory macrophages and reside near enteric neurons, we postulated that macrophage-derived C1q might also regulate enteric nervous system function.

As an initial test of this idea, we compared the numbers of enteric neurons in *C1qa*^ΔMφ and *C1qa*^fl/fl mice. Immunofluorescence analysis of LMMP wholemounts from the small intestine and colon revealed a similar number of HuC/D^+ neurons and a similar density of TUBB3^+ neuronal fibers (*Figure 5A and B*). There were also similar numbers of specific neuronal subsets, including excitatory (*Chat*^+) and inhibitory (*Nos1*^+) neurons (*Figure 5C and E*), and a similar density of S100B^+ enteric glial cells (*Figure 5D and E*). Thus, the anatomical features of the enteric nervous system are not appreciably altered in *C1qa*^ΔMφ mice.

## *C1qa*^ΔMφ mice have altered gastrointestinal motility

We next assessed whether *C1qa*^ΔMφ mice show evidence of altered neuronal function. We performed RNAseq on the colonic LMMP from *C1qa*^fl/fl and *C1qa*^ΔMφ littermates and then conducted unbiased

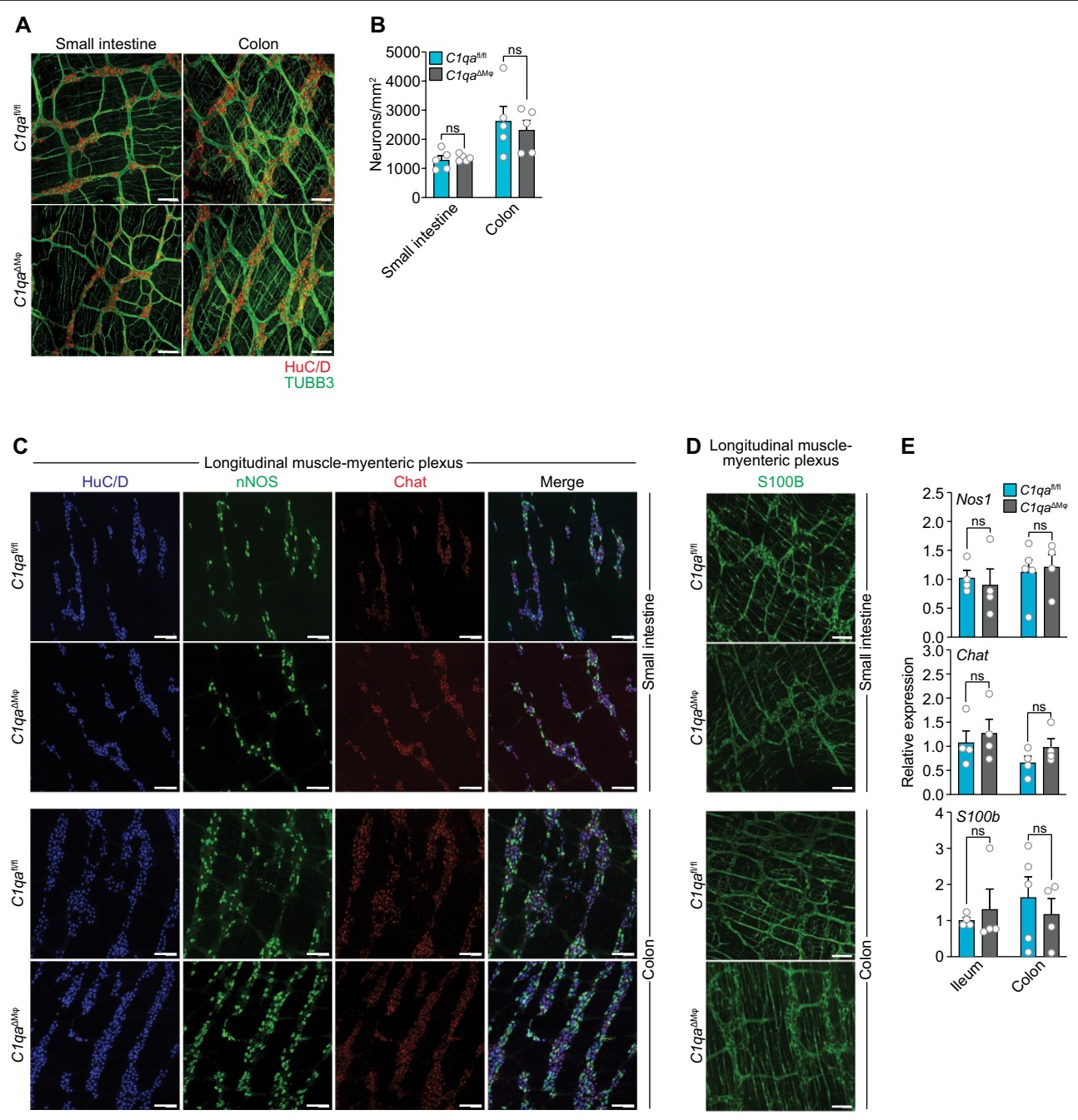

**Figure 5.** Numbers of enteric neurons are similar in *C1qa*^fl/fl^ and *C1qa*^ΔMφ^ mice. (**A**) Immunofluorescence analysis of enteric neuronal ganglia marked with HuC/D (red) and neuronal fibers marked with TUBB3 (green) in LMMP wholemounts of small intestines and colons from *C1qa*^fl/fl^ and *C1qa*^ΔMφ^ mice. Anti-mouse IgG AlexaFluor 594 and anti-rabbit IgG AlexaFluor 488 were used as secondary stains for HuC/D and TUBB3, respectively. Images are representative of three independent experiments. Scale bars = 50 μm. (**B**) Quantification of total enteric neurons per unit area (mm²) from the images shown in panel (**A**). Data are pooled from two independent experiments. Each data point represents one mouse. (**C**) Visualization of specific neuronal subsets in the LMMP from *C1qa*^fl/fl^ and *C1qa*^ΔMφ^ mice by RNAscope detection. Inhibitory neurons were marked by *Nos1* (green) and excitatory neurons were marked by *Chat* (red). Neuronal nuclei marked by HuC/D (blue) were detected by immunofluorescence. Images are representative of two independent experiments. Scale bars = 50 μm. (**D**) Immunofluorescence detection of enteric glial cells marked by S100B (green) in LMMP wholemounts from the small intestines and colons of *C1qa*^fl/fl^ and *C1qa*^ΔMφ^ mice. Images are representative of two independent experiments. Scale bars = 50 μm. (**E**) qPCR analysis of *Nos1*, *Chat*, and *S100b* in the LMMP of small intestines and colons from *C1qa*^fl/fl^ and *C1qa*^ΔMφ^ mice. Each data point represents one mouse. Error bars represent SEM. ns, not significant by the two-tailed Student's *t*-test. LMMP, longitudinal muscle-myenteric plexus.

Gene Set Enrichment Analysis. Of the 22 biological pathways that were enriched in the LMMP of $C1qa^{\Delta M\phi}$ mice, 17 were related to neuronal development or function, including synapse organization, dendrite development, and neurotransmitter secretion (*Figure 6A*). Our analysis also identified 30 differentially expressed genes with known roles in regulating neuronal activity (e.g. *Dusp26*), synaptic transmission (e.g. *Rasgrf2*), and neuropeptide signaling (e.g. *Tacr2*) (*Mao et al., 2017*; *Schwechter et al., 2013*; *Yang et al., 2017*; *Figure 6B*). We also compared the genes differentially expressed in the $C1qa^{\Delta M\phi}$ mice to those differentially expressed in the TashT mouse line, which contains an insertional mutation that leads to dysregulated gut motility. The gut motility phenotypes in the TashT line are comparable to Hirschsprung's disease, a human genetic disorder resulting in incomplete development of the enteric nervous system (*Bergeron et al., 2015*). A comparative analysis revealed a statistically significant overlap in the transcriptional changes in the colonic LMMP of $C1qa^{\Delta M\phi}$ mice and the neural crest cells of TashT mice (*Figure 6B*). These results suggested that macrophage C1q impacts enteric nervous system gene expression and function.

Efficient coordination of gastrointestinal motility is necessary for proper digestion, nutrient absorption, and excretion. Given that muscularis macrophages regulate enteric nervous system functions that govern gastrointestinal motility (*Muller et al., 2014*), we assessed whether macrophage C1q impacts gut motility. We first tested this idea by measuring gut transit time using the nonabsorbable dye Carmine Red. $C1qa^{\Delta M\phi}$ and $C1qa^{fl/fl}$ littermates were gavaged with the dye and the time to the first appearance of the dye in the feces was recorded. Transit times were decreased in $C1qa^{\Delta M\phi}$ mice relative to their $C1qa^{fl/fl}$ littermates, indicating accelerated gut motility (*Figure 6C*). This was not due to a change in the length of either the small intestine or the colon, which were unaltered in the $C1qa^{\Delta M\phi}$ mice (*Figure 6D*). By contrast, gut transit time was unchanged in $C3^{-/-}$ mice, suggesting that macrophage C1q impacts gut motility independent of its canonical function in the classical complement pathway (*Figure 6C*). Accelerated transit was also observed in the small intestines of $C1qa^{\Delta M\phi}$ mice as assessed by rhodamine dye transit assay (*Figure 6E*). To assess colonic motility, we measured the expulsion time after intrarectal insertion of a glass bead and found that $C1qa^{\Delta M\phi}$ mice had accelerated colonic motility when compared to $C1qa^{fl/fl}$ littermates (*Figure 6F*). Our results thus suggest that the absence of macrophage C1q results in defective enteric nervous system function and dysregulated gastrointestinal motility.

A limitation of in vivo measures of gut motility is that they cannot distinguish between defects in 'intrinsic' enteric neurons and 'extrinsic' neurons that innervate the gastrointestinal tract (*Berthoud et al., 2004*; *Uesaka et al., 2016*). We, therefore, used an ex vivo organ bath system to specifically assess enteric nervous system function by measuring the activity of colonic migrating motor complexes (CMMC; rhythmic peristaltic contractions that depend on the enteric nervous system) (*Obata et al., 2020*). Spatiotemporal mapping revealed that the colons of $C1qa^{\Delta M\phi}$ mice had increased total number, frequency, and velocity of CMMC as compared to $C1qa^{fl/fl}$ littermates (*Figure 6G and H*; *Figure 6—video 1*; *Figure 6—video 2*). This indicated that the colons of $C1qa^{\Delta M\phi}$ mice maintained increased neurogenic peristaltic activity compared to their $C1qa^{fl/fl}$ littermates even in the absence of gut-extrinsic signals. Thus, the absence of macrophage C1q increases enteric nervous system-dependent peristalsis and accelerates gut transit. Taken together, our findings reveal that macrophage C1q regulates gastrointestinal motility.

## Discussion

Here, we have identified a role for C1q in regulating gastrointestinal motility. We discovered that macrophages are the primary source of C1q in the mouse intestine and that macrophage C1q regulates enteric neuronal gene expression and gastrointestinal transit time. Our findings reveal a previously unappreciated function for C1q in the intestine and help to illuminate the molecular basis for macrophage-mediated control of gut motility.

Our study identifies macrophages as the main source of C1q in the mouse small intestine and colon. Both transient antibody-mediated depletion of macrophages and in vivo deletion of the *C1qa* gene from macrophages led to a marked reduction in intestinal *C1q* expression. The $C1qa^{\Delta M\phi}$ mice also lacked C1q in the circulation, indicating that LysM+ macrophages or macrophage-like cells are the sources of circulating C1q in the absence of infection. This enhances findings from prior studies indicating that monocytes, macrophages, and immature dendritic cells are the main sources of C1q in the bloodstream (*El-Shamy et al., 2018*). Importantly, the $C1qa^{\Delta M\phi}$ mice retained C1q expression

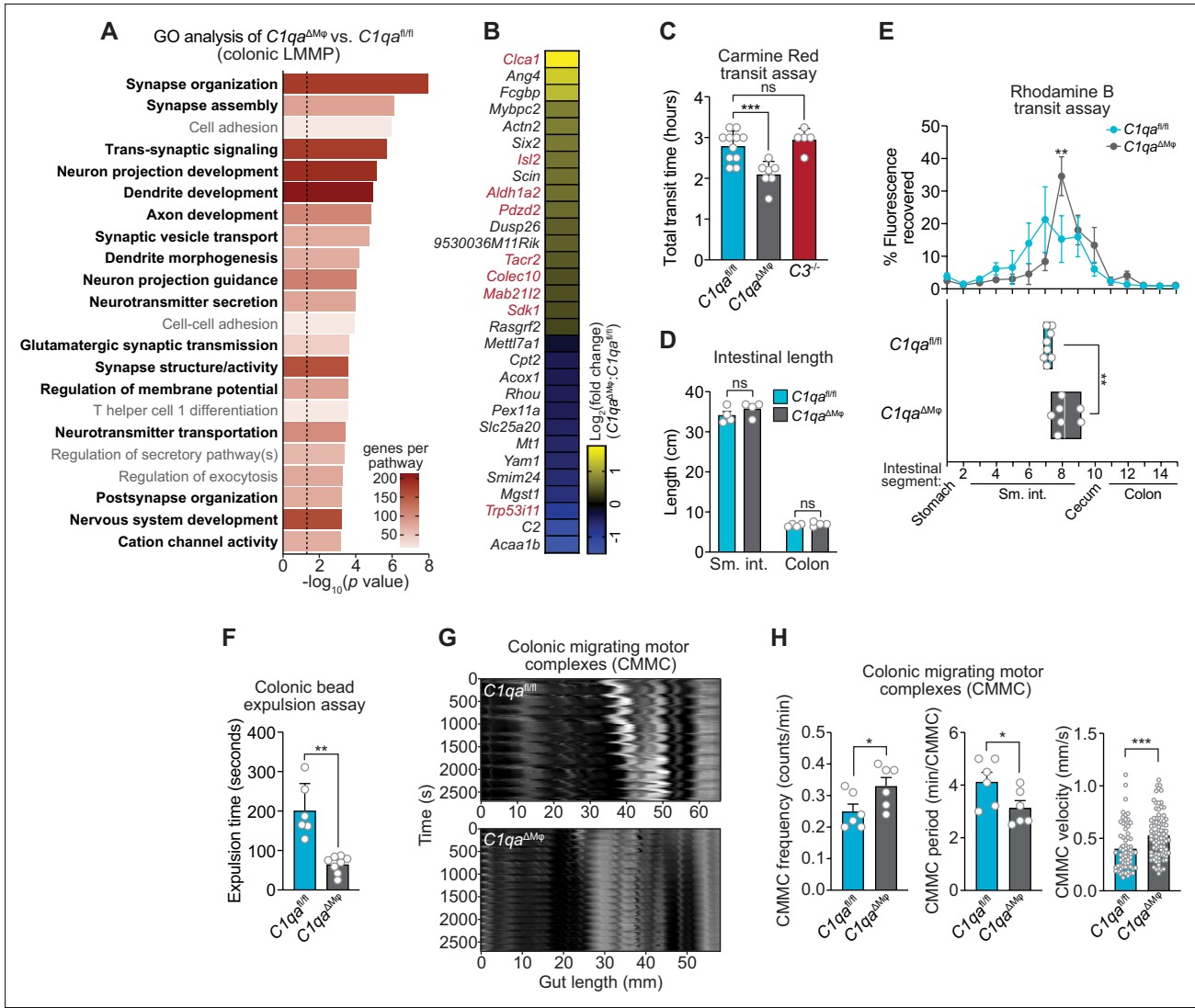

**Figure 6.** $C1qa^{\Delta M\varphi}$ mice have altered gastrointestinal motility. (**A**) RNA-seq was performed on colonic LMMP from $C1qa^{\Delta M\varphi}$ and $C1qa^{fl/fl}$ littermates. Annotated gene ontology (GO) biological processes were assigned to genes that were differentially expressed in $C1qa^{\Delta M\varphi}$ mice when compared to their $C1qa^{fl/fl}$ littermates. GO biological processes associated with neurons are in bold type. The dotted line indicates the cutoff for statistical significance. Five mice per group were analyzed as pooled biological replicates. Data are available from the Sequence Read Archive under BioProject ID PRJNA793870. (**B**) The colonic longitudinal muscle myenteric plexus of $C1qa^{\Delta M\varphi}$ mice have a transcriptional profile like that of mice with a gastrointestinal motility disorder. RNA-seq was performed on the colonic longitudinal muscle-myenteric plexus from five $C1qa^{fl/fl}$ and five $C1qa^{\Delta M\varphi}$ littermates. Genes that were differentially expressed are represented in a heatmap that depicts $\log_2$(fold change). Genes that also showed altered expression in the TashT mouse line, which is a model of human Hirschsprung's disease (*Bergeron et al., 2015*), are indicated in red. Statistical significance of the overlap between differentially expressed genes in $C1qa^{\Delta M\varphi}$ and TashT mice was determined by Fisher's exact test (p=0.0032). (**C**) Measurement of total intestinal transit time in $C1qa^{fl/fl}$ and $C1qa^{\Delta M\varphi}$ littermates and $C3^{-/-}$ mice. Mice were gavaged with 100 µl of Carmine Red [5% (w/v in 1.5% methylcellulose)]. Fecal pellets were collected every 15 min and transit time was recorded when the dye was first observed in the feces. Each data point represents one mouse and the results are pooled from five independent experiments. (**D**) Intestinal tract length is not altered in $C1qa^{\Delta M\varphi}$ mice. Small intestines and colons from $C1qa^{fl/fl}$ and $C1qa^{\Delta M\varphi}$ littermates were excised and measured. Each data point represents one mouse. (**E**) Transit of rhodamine B-dextran through the intestines of $C1qa^{fl/fl}$ and $C1qa^{\Delta M\varphi}$ littermates. Mice were sacrificed 90 min after gavage with rhodamine B-dextran. The intestines were divided into 16 segments, the rhodamine B fluorescence was measured in each segment (top panel), and the geometric center of the fluorescence was determined for each mouse (bottom panel). Each data point represents one mouse and the results were pooled from four independent experiments. (**F**) Colonic motility was measured by determining the expulsion time of a glass bead inserted intrarectally into $C1qa^{fl/fl}$ and $C1qa^{\Delta M\varphi}$ littermates. Each data point represents one mouse and the results are representative of three independent experiments. (**G**) Representative spatiotemporal maps of colonic migrating motor complex (CMMC) formation in colons of $C1qa^{fl/fl}$ and $C1qa^{\Delta M\varphi}$ mice. Representative video recordings were captured in *Figure 6—video 1* ($C1qa^{fl/fl}$ mice) and *Figure 6—video 2* ($C1qa^{\Delta M\varphi}$ mice). Each map represents one mouse and is representative of two independent experiments. (**H**) Analysis of CMMC parameters in colons of $C1qa^{fl/fl}$ and $C1qa^{\Delta M\varphi}$ mice. Each data point represents one mouse (for CMMC frequency and CMMC period)

*Figure 6 continued on next page*

*Figure 6 continued*

or one individual CMMC event (for velocity). Data are pooled from two independent experiments. LMMP, longitudinal muscle-myenteric plexus; sm. int., small intestine. Error bars represent SEM. *p<0.05; **p<0.01; ***p<0.001; ****p<0.0001; ns, not significant by the two-tailed Student's *t*-test.

The online version of this article includes the following video and figure supplement(s) for figure 6:

**Figure supplement 1.** Single-cell RNA-seq analysis of intestinal macrophages from *C1qa*$^{\Delta M\varphi}$ and *C1qa*$^{fl/fl}$ littermates.

**Figure supplement 2.** The gene encoding complement component 1q (C1q) receptor BAI1 (*Adgrb1*) is expressed by enteric neurons.

**Figure 6—video 1.** Ex vivo recording of colonic peristalsis in *C1qa*$^{fl/fl}$ mice.

https://elifesciences.org/articles/78558/figures#fig6video1

**Figure 6—video 2.** Ex vivo recording of colonic peristalsis in *C1qa*$^{\Delta M\varphi}$ mice.

https://elifesciences.org/articles/78558/figures#fig6video2

in the brain, allowing us to analyze the effects of C1q deficiency without possible confounding effects on the central nervous system.

C1q has two known physiological functions that are distinct and vary according to tissue context. C1q was originally discovered as having a role in the classical complement pathway, which tags and destroys invading microbes (*Noris and Remuzzi, 2013*; *Schifferli et al., 1986*). Circulating C1q binds to invading microorganisms and recruits additional proteins that assemble into the membrane attack complex (MAC) (*Kishore and Reid, 2000*). C1q-mediated MAC formation has been described primarily in the bloodstream, where the necessary accessory proteins are present at high levels (*Davis et al., 1979*). However, even in the absence of infection, C1q is expressed in tissues such as the brain, where it regulates neuronal development and function (*Kouser et al., 2015*; *van Schaarenburg et al., 2016*).

Our findings suggest that C1q does not play a central role in the immune defense of the intestine. First, we found that intestinal C1q expression was not induced by gut commensals or pathogens and was not deposited into the gut lumen. Second, C1q deficiency did not markedly alter gut microbiota composition or the course of disease after DSS treatment. There were also no major changes in cytokine expression or numbers and frequencies of intestinal immune cells that would indicate dysregulated interactions with the microbiota. Third, C1q was not required for clearance of *C. rodentium*, a non-disseminating enteric pathogen whose clearance requires antigen-specific IgG and complement component 3 (C3) (*Belzer et al., 2011*). Although we cannot rule out a role for C1q in immune defense against other intestinal pathogens, or during chronic inflammation or infection, these findings suggest that C1q is not essential for intestinal immune defense in mice.

Instead, our results indicate that C1q influences enteric nervous system function and regulates intestinal motility. First, C1q-expressing macrophages were present in the myenteric and submucosal plexuses and resided close to enteric neurons. Second C1q-expressing macrophages expressed cell surface markers like those expressed by nerve-adjacent C1q-expressing macrophages in the lung (*Ural et al., 2020*). Third, macrophage-specific deletion of *C1qa* altered enteric neuronal gene expression. Finally, consistent with the altered neuronal gene expression, macrophage-specific *C1qa* deletion altered gastrointestinal motility in both the small and large intestines. Thus, our results suggest that the function of C1q in the intestine is similar to its function in the brain, where it regulates the development and function of neurons (*Benoit and Tenner, 2011*; *Kouser et al., 2015*; *van Schaarenburg et al., 2016*).

A function for macrophage C1q in intestinal motility adds to the growing understanding of how gut macrophages regulate intestinal peristalsis. Prior work has shown that CSF1R$^+$ macrophages selectively localize to the muscularis of the mouse intestine (*Muller et al., 2014*; *Gabanyi et al., 2016*). These macrophages secrete BMP2, which activates enteric neurons that regulate colonic muscle contraction and thus colonic motility (*Muller et al., 2014*). We found that depletion of CSF1R$^+$ macrophages reduces intestinal C1q expression and that macrophage-specific deletion of *C1qa* alters enteric neuronal gene expression and activity. Thus, our findings suggest that C1q is a key component of the macrophage-enteric nervous system axis.

An important remaining question concerns the molecular mechanism by which C1q regulates gut motility. One possibility is that C1q shapes microbiota composition which, in turn, impacts gut motility. This idea is suggested by studies in zebrafish showing that a deficiency in intestinal macrophages leads to altered gut microbiota composition relative to wild-type zebrafish (*Earley et al., 2018*). Other

studies in zebrafish and mice have shown that severe defects in enteric nervous system development produce changes in gut microbiota composition that are linked to dysregulated gut motility (*Rolig et al., 2017*; *Johnson et al., 2018*). However, we did not observe prominent changes in the composition of the gut microbiota in *C1qa*^ΔMϕ mice, arguing against a central role for the microbiota in C1q-mediated regulation of gut motility. A second possibility is that the absence of C1q leads to immunological defects that alter gut transit time. This idea is consistent with studies showing that T-cell cytokines can influence gastrointestinal motility (*Akiho et al., 2011*). However, this seems unlikely given the lack of pronounced immunological abnormalities in the intestines of *C1qa*^ΔMϕ mice.

A third possibility is that C1q changes the cell-intrinsic properties of the macrophages that express it, thus altering their interactions with neurons to influence gut motility. We explored this possibility by conducting single-cell RNA sequencing (scRNAseq) on macrophages isolated from small intestinal cell suspensions (*Figure 6—figure supplement 1A*). We identified 11 unique macrophage clusters and found that *C1qa*^ΔMϕ mice had alterations in at least three highly represented clusters (*Figure 6—figure supplement 1B*). Gene set enrichment analysis of the most significantly altered clusters did not reveal any pronounced functional differences (*Figure 6—figure supplement 1C*). However, analysis of the differentially expressed genes across all macrophage clusters indicated lowered representation of transcripts that are linked to control of macrophage differentiation or functional states, such as *Malat1*, *Neat1*, and *Etv3* (*Cui et al., 2019*; *Gao et al., 2020*; *Villar et al., 2023*; *Zhang et al., 2019*; *Figure 6—figure supplement 1D*). Furthermore, a recent study identified a set of 13 'microglia-specific genes' that represent a unique transcriptional overlap between microglia in the CNS and intestinal macrophages (*Verheijden et al., 2015*). In macrophages from *C1qa*^fl/fl mice, we observed the expression of eight 'microglia-specific genes' whose expression was lowered or lost in macrophages from *C1qa*^ΔMϕ mice (*Figure 6—figure supplement 1E*). Thus, it is possible that altered intestinal motility could arise in part from cell-intrinsic functional alterations in C1q-deficient intestinal macrophages. Such alterations could arise from a C1q autocrine signaling loop or C1q could imprint a neuronal function that feeds back to regulate macrophage gene expression as exemplified in *Muller et al., 2014*.

A fourth possibility is that C1q^+ macrophages engulf specific neurons. Indeed, macrophages restrain neurogenesis in the enteric nervous system through phagocytosis of apoptotic neurons, which is consistent with the ability of C1q to opsonize dying host cells (*Kulkarni et al., 2017*; *Botto et al., 1998*; *Korb and Ahearn, 1997*). However, we observed no marked differences in the overall numbers of enteric neurons or numbers of excitatory and inhibitory neurons when comparing *C1qa*^ΔMϕ and *C1qa*^fl/fl mice, which argues against this possibility. A fifth possibility is that C1q acts directly on enteric smooth muscle cells that regulate gut motility. Although we cannot rule out this possibility, our transcriptional profile of the colonic myenteric plexus of *C1qa*^ΔMϕ mice suggests that most of the transcriptional changes were associated with neuronal function and homeostasis.

Given that the *C1qa*^ΔMϕ mice showed altered neuronal gene expression, a sixth possibility is that C1q interacts directly with enteric neurons or glial cells as a signaling molecule. Like macrophage-produced BMP2 (*Muller et al., 2014*), C1q might bind to specific receptors on neurons to regulate their activity. In support of this idea, we observed that mouse enteric neurons express *Adgrb1*, which encodes BAI1 (*Figure 6—figure supplement 2A and B*), a recently identified C1q receptor on human neural stem cells (*Benavente et al., 2020*). These data suggest a possible signaling axis for C1q-mediated control of enteric nervous system function.

Our findings on intestinal C1q have implications for human intestinal disease. Indeed, single-cell RNAseq analysis shows that macrophages recovered from the human intestinal muscularis selectively express *C1q* when compared to lamina propria macrophages (*Domanska et al., 2022*). Dysregulated peristalsis is a characteristic of irritable bowel syndrome (*Vrees et al., 2002*) and is present in a subset of inflammatory bowel disease patients (*Bassotti et al., 2014*). Our finding that macrophage C1q regulates gut motility could suggest new strategies to prevent or treat these diseases. Additionally, most humans with C1q deficiency develop systemic lupus erythematosus (SLE). Since C1q can target cellular debris for phagocytosis, it is thought that C1q deficiency results in increased exposure of self-antigen to the immune system, thereby reducing immune tolerance and causing autoimmune disease (*Macedo and Isaac, 2016*). Furthermore, roughly 42.5% of SLE patients report gastrointestinal symptoms that range from acute abdominal pain to chronic intestinal obstruction (*Fawzy et al., 2016*; *Tian and Zhang, 2010*). The exact cause of these symptoms

is unclear. Given that C1q deficiency is strongly correlated with SLE in humans and alters gut motility in mice, we suggest that C1q could be a therapeutic target for SLE patients that present with chronic constipation or other forms of dysregulated intestinal motility.

# Materials and methods

## Key resources table

| Reagent type (species) or resource | Designation | Source or reference | Identifiers | Additional information |
|---|---|---|---|---|
| Strain, strain background (*Mus musculus*) | *C1qa*^fl/fl^; B6(SJL)-C1qa^tm1c(EUCOMM)Wtsi^/TennJ | Jackson Laboratory; *Fonseca et al., 2017* | Stock #031261 | |
| Strain, strain background (*Mus musculus*) | LysM-Cre; B6.129P2-Lyz2^tm1(cre)Ifo^/J | Jackson Laboratory; *Clausen et al., 1999* | Stock #004781 | |
| Strain, strain background (*Mus musculus*) | *C1qa*^ΔMΦ^ | this paper | | Generated by crossing *C1qa*^fl/fl^ mice with LysM-Cre mice |
| Strain, strain background (*Mus musculus*) | *C3*^-/-^; B6.129S4-C3^tm1Crr^/J | Jackson Laboratory; *Wessels et al., 1995* | Stock #029661 | |
| Strain, strain background (*Mus musculus*) | Germ-free C57BL/6 J mice | UT Southwestern Gnotobiotics Core Facility | | |
| Strain, strain background (*Salmonella enterica*) | *Salmonella enterica* subsp. enterica serovar Typhimurium strain SL1344 | Dr. Vanessa Sperandio; *Eichelberg and Galán, 1999* | | |
| Strain, strain background (*Citrobacter rodentium*) | *Citrobacter rodentium* strain DBS100 | ATCC | Strain# 51459 | |
| Antibody | Anti-Actin HRP (rabbit monoclonal) | Cell Signaling | Clone: 13E5 | Immunoblot (1:5000) |
| Antibody | Anti-ARG1 (sheep monoclonal) | R&D Systems | Clone: P05089 | Flow (1:100) |
| Antibody | Anti-B220 (rat monoclonal) | Thermo Fisher | Clone: RA3-6B2 | Flow (1:500) |
| Antibody | Anti-C1q (rat monoclonal) | Cedarlane Laboratories | Clone: RmC7H8 | Flow (1:50) |
| Antibody | Anti-C1q (rabbit polyclonal) | Thermo Fisher | Cat# PA5-29586 | Immunoblot (1:500) |
| Antibody | Anti-C1q-biotin (mouse monoclonal) | Abcam | Clone: JL1 | ELISA (1:1000); Immunofluorescence (1:100) |
| Antibody | Anti-CD3 (rat monoclonal) | Thermo Fisher | Clone: 17A2 | Flow (1:200) |
| Antibody | Anti-CD4 (rat monoclonal) | BioLegend | Clone: GK1.5 | Flow (1:500) |
| Antibody | Anti-CD11b (rat monoclonal) | Thermo Fisher | Clone: M1/70 | Flow (1:200) |
| Antibody | Anti-CD11c (Armenian hamster monoclonal) | Thermo Fisher | Clone: N418 | Flow (1:500) |
| Antibody | Anti-CD16/32 (rat monoclonal) | BioLegend | Clone: 93 | Fc receptor block (1:1000) |
| Antibody | Anti-CD19 (rat monoclonal) | BioLegend | Clone: 1D3 | Flow (1:500) |
| Antibody | Anti-CD45 (rat monoclonal) | BioLegend | Clone: 30-F11 | Flow (1:500) |
| Antibody | Anti-CD90.2 (rat monoclonal) | BioLegend | Clone: 30-H12 | Flow (1:500) |
| Antibody | Anti-CD169 (rat monoclonal) | BioLegend | Clone: 3D6.112 | Flow (1:200) |

*Continued on next page*

*Continued*

| Reagent type (species) or resource | Designation | Source or reference | Identifiers | Additional information |
|---|---|---|---|---|
| Antibody | Anti-CD169 (rat monoclonal) | Abcam | Clone: 3D6.112 | Immunofluorescence (1:200) |
| Antibody | Anti-CSF1R (rat monoclonal) | Bio X Cell | Cat# AFS98 | Macrophage depletion (100 mg/kg) |
| Antibody | Anti-F4/80 (rat monoclonal) | BioLegend | Clone: BM8 | Flow (1:100) |
| Antibody | Anti-FoxP3 (rat monoclonal) | Thermo Fisher | Clone: FJK-16s | Flow (1:50) |
| Antibody | Anti-GATA3 (mouse monoclonal) | BD Biosciences | Clone: L50-823 | Flow (1:50) |
| Antibody | Anti-IgA (rat monoclonal) | Thermo Fisher | Clone: 11-44-2 | Flow (1:50) |
| Antibody | Anti-LY6C (rat monoclonal) | BioLegend | Clone: RB6-8C5 | Flow (1:500) |
| Antibody | Anti-MHCII (rat monoclonal) | Thermo | Clone: M5/114.15.2 | Flow (1:500) |
| Antibody | Anti-REG3G antiserum (rabbit polyclonal) | *Cash et al., 2006*; antiserum generated by Pacific Biosciences | | Immunoblot (1:1000) |
| Antibody | Anti-RORγt (rat monoclonal) | Thermo Fisher | Clone: AFKJS-9 | Flow (1:50) |
| Antibody | Anti-T-BET (mouse monoclonal) | BioLegend | Clone: 4B10 | Flow (1:50) |
| Antibody | Anti-TREM2 (rat monoclonal) | R&D Systems | Clone: 237920 | Flow (1:200) |
| Antibody | Anti-TUBB3 (rabbit polyclonal) | Abcam | Cat# ab18207 | Immunofluorescence (1:200) |
| Antibody | Anti-S100β (rabbit polyclonal) | Dako | Cat# GA504 | Immunofluorescence |
| Antibody | Anti-HuC/D (rabbit monoclonal) | Abcam | Cat# ab184267 | Immunofluorescence (1:400) |
| Antibody | Goat anti-rabbit IgG HRP conjugate | Abcam | Cat# ab6721 | Immunoblot (1:5000) |
| Antibody | secondary antibodies – donkey polyclonal anti-rabbit/rat/mouse AlexaFluor 488/594/647 | Invitrogen | | Immunofluorescence (1:400) |
| Antibody | mouse IgG1 | Abcam | Cat# ab18443 | ELISA (10 µg/ml) |
| Antibody | Rat IgG2a | Thermo Fisher | Clone: 2A3 | Isotype control for anti-CSF1R macrophage depletion (100 mg/kg) |
| Antibody | Rat IgG1 PE isotype control | Cedarlane Laboratories | Cat# CLCR104 | Flow (1:50) |
| Sequence-based reagent | mouse *C1qa* TaqMan assay | Thermo Fisher | Assay ID: Mm00432142_m1 | |
| Sequence-based reagent | mouse *C1qb* TaqMan assay | Thermo Fisher | Assay ID: Mm01179619_m1 | |
| Sequence-based reagent | mouse *C1qc* TaqMan assay | Thermo Fisher | Assay ID: Mm00776126_m1 | |
| Sequence-based reagent | mouse *Chat* TaqMan assay | Thermo Fisher | Assay ID: Mm01221880_m1 | |
| Sequence-based reagent | mouse *Nos1* TaqMan assay | Thermo Fisher | Assay ID: Mm01208059_m1 | |
| Sequence-based reagent | mouse *S100b* TaqMan assay | Thermo Fisher | Assay ID: Mm00485897_m1 | |
| Sequence-based reagent | mouse *Reg3g* TaqMan assay | Thermo Fisher | Assay ID: Mm00441127_m1 | |
| Sequence-based reagent | mouse *Ifng* TaqMan assay | Thermo Fisher | Assay ID: Mm01168134_m1 | |
| Sequence-based reagent | mouse *Il4* TaqMan assay | Thermo Fisher | Assay ID: Mm00445259_m1 | |

*Continued on next page*

*Continued*

| Reagent type (species) or resource | Designation | Source or reference | Identifiers | Additional information |
|---|---|---|---|---|
| Sequence-based reagent | mouse *IL5* TaqMan assay | Thermo Fisher | Assay ID: Mm00439646_m1 | |
| Sequence-based reagent | mouse *Il10* TaqMan assay | Thermo Fisher | Assay ID: Mm01288386_m1 | |
| Sequence-based reagent | mouse *Il13* TaqMan assay | Thermo Fisher | Assay ID: Mm00434204_m1 | |
| Sequence-based reagent | mouse *Il17a* TaqMan assay | Thermo Fisher | Assay ID: Mm00439618_m1 | |
| Sequence-based reagent | mouse *Il17f* TaqMan assay | Thermo Fisher | Assay ID: Mm00521423_m1 | |
| Sequence-based reagent | mouse 18 S gene TaqMan assay | Thermo Fisher | Assay ID: Mm03928990_g1 | |
| Sequence-based reagent | bacterial 16 S universal rRNA forward primer | Gift from Dr. Andrew Koh | | 5'- ACTCCTACGGGAGGCAGCAGT-3' |
| Sequence-based reagent | Bacterial 16 S universal rRNA reverse primer | Gift from Dr. Andrew Koh | | 5'- ATTACCGCGGCTGCTGGC-3' |
| Sequence-based reagent | bacterial 16 S V3 - rRNA gene forward primer | Thermo Fisher; (*Klindworth et al., 2013*) | 16 S rRNA gene sequencing | 5'-TCGTCGGCAGCGTCAGATGTGTATAAGAGAC AGCCTACGGGNGGCWGCAG-3' |
| Sequence-based reagent | bacterial 16 S v4 - rRNA gene reverse primer | Thermo Fisher; *Klindworth et al., 2013* | 16 S rRNA gene sequencing | 5'- GTCTCGTGGGCTCGGAGATGTGTA TAAGAGACAGGACTACHVGGGTATCTAATCC-3' |
| Sequence-based reagent | mouse *C1qa* RNAscope probe (C1) | Advanced Cell Diagnostics | Cat# 498241 | |
| Sequence-based reagent | mouse *C1qa* RNAscope probe (C3) | Advanced Cell Diagnostics | Cat# 498241-C3 | |
| Sequence-based reagent | mouse *Chat* RNAscope probe (C1) | Advanced Cell Diagnostics | Cat# 408731 | |
| Sequence-based reagent | mouse *Nos1* RNAscope probe (C2) | Advanced Cell Diagnostics | Cat# 437651-C2 | |
| Sequence-based reagent | mouse *Adgrb1* RNAscope probe (C1) | Advanced Cell Diagnostics | Cat# 317901 | |
| Sequence-based reagent | mouse *Csf1r* RNAscope probe (C2) | Advanced Cell Diagnostics | Cat# 428191-C2 | |
| Peptide, recombinant protein | recombinant mouse C1q | Complementech | Cat# M099 | |
| Commercial assay or kit | Chromium Next GEM Single Cell 3' Kit v3.1 | 10 x Genomics | Cat# PN-1000269 | |
| Commercial assay or kit | Chromiium Next GEM Chip G Single Cel Kit | 10 x Genomics | Cat# PN-1000127 | |
| Commercial assay or kit | Dual Index Kit TT Set A | 10 x Genomics | Cat# PN-1000215 | |
| Commercial assay or kit | FOXP3/Transcription Factor Fixation/Permeabilization Buffer Set | Thermo Fisher | Cat# 00-5523-00 | |

*Continued on next page*

*Continued*

| Reagent type (species) or resource | Designation | Source or reference | Identifiers | Additional information |
|---|---|---|---|---|
| Commercial assay or kit | MMLV Reverse Transcriptase Kit | Thermo Fisher | Cat# 28025–021 | |
| Commercial assay or kit | NextSeq 500/550 High Output Kit v2.5 | Illumina | Cat# 20024907 | |
| Commercial assay or kit | PE300 (Paired end 300 bp) v3 kit | Illumina | Cat# MS-102–3001 | |
| commercial assay or kit | RNAscope Fluorescent Multiple Reagent Kit | Advanced Cell Diagnostics | Cat# 320850 | |
| Commercial assay or kit | RNeasy Universal Mini Kit | Qiagen | Cat# 73404 | |
| Commercial assay or kit | DNEasy Blood & Tissue Kit | Qiagen | Cat# 69504 | |
| Commercial assay or kit | TaqMan Master Mix | Thermo Fisher | Cat# 4369542 | |
| Commercial assay or kit | TruSeq RNA sample preparation kit | Illumina | Cat# RS-122–2001 | |
| Commercial assay or kit | SsoAdvanced Universal SYBR Green Supermix | BioRad | Cat# 1725270 | |
| Chemical compound, drug | Agencourt AmpureXP beads | Beckman Coulter Genomics | Cat# A63880 | |
| Chemical compound, drug | Carmine Red | Sigma | Cat# C1022-25G | |
| Chemical compound, drug | Collagenase IV | Sigma | Cat# C5138-1G | |
| Chemical compound, drug | Borosilicate glass beads (2 mm) | Millipore Sigma | Cat# Z273627-1EA | |
| Chemical compound, drug | Dextran sulfate sodium | Thomas Scientific | Cat# 216011090 | |
| Chemical compound, drug | DNase I | Sigma | Cat# DN25 | |
| Chemical compound, drug | Dispase II | Sigma | Cat# D4693-1G | |
| Chemical compound, drug | FITC-dextran (4000 Da) | Sigma | Cat# FD4-1g | |
| Chemical compound, drug | Ghost 710 | Tonbo Biosciences | Cat# 13–0871 T100 | Flow cytometry viability dye |
| Chemical compound, drug | Methylcellulose | Sigma | Cat# M0262-100G | |
| Chemical compound, drug | Nalidixic acid, sodium salt | Research Products International | Cat# N23100-25.0 | |

*Continued on next page*

*Continued*

| Reagent type (species) or resource | Designation | Source or reference | Identifiers | Additional information |
|---|---|---|---|---|
| Chemical compound, drug | Optimal Cutting Temperature Compound (OCT) | Thermo Fisher | Cat# 23-730-571 | |
| Chemical compound, drug | Percoll Plus | GE Healthcare | Cat# GE17-0891-09 | |
| Chemical compound, drug | 4% Paraformaldehyde Solution | Thermo Fisher | Cat# J19943.K2 | |
| Chemical compound, drug | Normal donkey serum | Southern Biotech | Cat# 0030–01 | |
| Chemical compound, drug | Triton X-100 | Thermo Fisher | Cat# A16046.AP | |
| Chemical compound, drug | Protease inhibitors | Millipore Sigma | Cat# 11836153001 | |
| Chemical compound, drug | Rhodamine B-dextran | Thermo Fisher | Cat# D1841 | |
| Chemical compound, drug | Streptavidin-Cy5 | Thermo Fisher | Cat# 434316 | |
| Chemical compound, drug | Streptavidin-HRP conjugate | Abcam | Cat# ab7403 | ELISA |
| Chemical compound, drug | Sylgard 184 Silicone Elastomer | Fisher Scientific | Cat# 4019862 | |
| Chemical compound, drug | VECTASHIELD Antifade Mounting Medium with 4',6-diamidino-2-phenylindole (DAPI) | Vector Labs | Cat# H-1200–10 | |
| Software, algorithm | Cell Ranger Single-Cell Software Suite | 10 X Genomics | | |
| Software, algorithm | clusterProfiler | *Yu et al., 2012* | | |
| Software, algorithm | CLC Genomics Workbench | Qiagen | | |
| Software, algorithm | CLC Bio microbial genomics module | Qiagen | | https://digitalinsights.qiagen.com/plugins/clc-microbial-genomics-module/ |
| Software, algorithm | FlowJo | BD Biosciences | | |
| Software, algorithm | ggplot2 | *Love et al., 2015* | | |
| Software, algorithm | GraphPad PRISM | GraphPad Software | Version 7.0; RRID:SCR_002798 | |
| Software, algorithm | Gut Analysis Toolbox | *Sorensen et al., 2022* | | |

*Continued*

| Reagent type (species) or resource | Designation | Source or reference | Identifiers | Additional information |
|---|---|---|---|---|
| Software, algorithm | Igor Pro 9 | WaveMetrics | | |
| Software, algorithm | Illumina Nextera Protocol | Illumina | Part # 15044223 Rev. B | |
| Software, algorithm | ImageJ | National Institutes of Health | | https://imagej.nih.gov/ij/ |
| Software, algorithm | *Limma* | *Ritchie et al., 2015* | | |
| Software, algorithm | NovoExpress | Agilent Technologies | | |
| Software, algorithm | PVCAM software | Teledyne Photometrics | | |
| Software, algorithm | Seurat V3 R Package | *Stuart et al., 2019* | | |
| Other | Agilent 2100 Bioanalyzer | Agilent Technologies | G2939A | RNA integrity analysis |
| Other | Amicon Ultra centrifugal filters | Millipore | Cat #UFC900324 | Fecal protein extraction |
| Other | BioRad ChemiDoc Touch System | BioRad | Cat# 1708370 | Western blot imaging: |
| Other | Chromium Controller & Next GEM Accessory Kit | 10 X Genomics | Cat# PN-120223 | Single cell RNA sequencing library construction |
| Other | CMOS camera | Teledyne Photometrics | MOMENT | Ex vivo peristalsis: |
| Other | Leica CM1950 (Cryostat) | Leica | | Cryosectioning |
| Other | FACSAria | BD Biosciences | | Flow cytometric cell sorting |
| Other | ORCA-Fusion sCMOS camera | Hamamatsu Photonics | C14440-20UP | Imaging |
| Other | Illumina MiSeq | Illumina | RRID:SCR_016379 | 16 S rRNA |
| Other | Illumina NextSeq 550 | Illumina | | Bulk RNA sequencing and single cell RNA sequencing |
| Other | Keyence Fluorescence Microscope | Keyence | BZ-X800 | Immunofluorescence |
| Other | NovoCyte 3005 | Agilent Technologies | | Flow cytometry analysis |
| Other | Organ bath chamber | Tokai Hit | | Ex vivo peristalsis |
| Other | Peristaltic pump | Gilson | MINIPULS3 | Ex vivo peristalsis |
| Other | QuantStudio 7 Flex Real-Time PCR System | Applied Biosystems | Cat #4485701 | qPCR analysis |
| Other | SpectraMax M5 plate reader | Molecular Devices | | ELISA and small intestinal motility analysis |
| Other | Zeiss Axio Imager M1 Microscope | Zeiss | | Immunofluorescence |

## Mice

Wild-type C57BL/6 J (Jackson Laboratory) and *C3*[-/-] mice (Jackson Laboratory; *Wessels et al., 1995*) were bred and maintained in the SPF barrier facility at the University of Texas Southwestern Medical Center. *C1qa*[ΔMφ] mice were generated by crossing *C1qa*[fl/fl] mice (Jackson Laboratory; *Fonseca et al., 2017*) with a mouse expressing Cre recombinase controlled by the macrophage-specific mouse *Lyz2* promoter (LysM-Cre mice; Jackson Laboratory; *Clausen et al., 1999*). Mice that were 8–12 weeks of age were used for all experiments and cohoused littermates were used as controls (i.e. Cre[+] and Cre[-] mice were from the same breeding pair). Both male and female mice were analyzed in experiments involving wild-type mice. Males were used for experiments involving *C1qa*[fl/fl] and *C1qa*[ΔMφ] mice.

Germ-free C57BL/6 J mice were bred and maintained in isolators at the University of Texas Southwestern Medical Center. All procedures were performed in accordance with protocols approved by the Institutional Animal Care and Use Committees (IACUC) of the UT Southwestern Medical Center.

## Quantitative polymerase chain reaction (qPCR)

Tissue RNA was isolated using the RNeasy Universal Mini kit (Qiagen, Hilden, Germany). Cellular RNA was isolated using the RNAqueous Micro kit (Thermo Fisher). cDNA was generated from the purified RNA using the M-MLV Reverse Transcriptase kit (Thermo Fisher). qPCR analysis was performed using TaqMan primer/probe sets and master mix (Thermo Fisher) on a Quant-Studio 7 Flex Real-Time PCR System (Applied Biosystems). Transcript abundances were normalized to 18 S rRNA abundance. TaqMan probe assay IDs are provided in the Key Resources table.

## Isolation and analysis of intestinal immune cells

Lamina propria cells were isolated from the intestine using a published protocol (*Yu et al., 2013*; *Yu et al., 2014*). Briefly, intestines were dissected from mice and Peyer's patches were removed. Intestines were cut into small pieces and thoroughly washed with ice-cold phosphate-buffered saline (PBS) containing 5% fetal bovine serum (PBS-FBS). Epithelial cells were removed by incubating intestinal tissues in Hank's buffered salt solution (HBSS) supplemented with 2 mM EDTA, followed by extensive washing with PBS-FBS. Residual tissues were digested twice with Collagenase IV (Sigma), DNase I (Sigma), and Dispase (BD Biosciences) for 45 min at 37 °C with agitation. Cells were filtered through 70 µm cell strainers (Thermo Fisher) and applied onto a 40%:80% Percoll gradient (GE Healthcare). Subepithelial cell populations were recovered at the interface of the 40% and 80% fractions. For small intestinal cell suspensions, the epithelial fraction was kept and combined with enzymatically liberated subepithelial cells. Cells were washed with 2 mM EDTA/3% FBS in PBS and Fc receptors were blocked with anti-CD16/32 (93). Cells were then stained with the viability dye Ghost 710 (Tonbo Biosciences) followed by antibodies against cell surface markers including anti-CD45 (30-F11), anti-CD11b (M1/70), anti-MHCII (M5/114.15.2), anti-F4/80 (BM8), anti-CD3 (17A2), anti-CD4 (GK1.5), anti-CD19 (1D3), anti-B220 (RA3-6B2), anti-CD11c (N418), anti-CD169 (3D6.112), anti-TREM2 (237920), and anti-LY6C (RB6-8C5). Cells were fixed and permeabilized with the eBioscience FOXP3/Transcription Factor Fixation/Permeabilization buffer set (Thermo Fisher) and then subjected to intracellular staining with anti-C1Q (RmC7H8), anti-FOXP3 (FJK-16s), anti-GATA3 (L50), anti-T-BET (4B10), anti-RORγ (AFKJS-9), and anti-ARG1 (P05089). Cells were sorted using a FACSAria (BD Biosciences) or analyzed using a NovoCyte 3005 (Agilent Technologies). Data were processed with FlowJo software (BD Biosciences) or NovoExpress (Agilent Technologies).

## Macrophage depletion

Anti-mouse CSF1R (Thermo Fisher; AFS98) and rat IgG2a isotype control (Thermo Fisher; 2A3) antibodies were administered intraperitoneally at a concentration of 100 mg/kg. Mice were sacrificed 72 hr post-injection and terminal ileum and colon were collected for qPCR analysis.

## Protein extraction from intestinal cells and feces

To isolate proteins from intestinal cell suspensions, cell pellets were resuspended in 100 µl of RIPA Lysis Buffer (Thermo Fisher) supplemented with protease inhibitors (Millipore Sigma) and vortexed vigorously every 5 min for 20 min. Lysates were cleared of cellular debris by centrifugation at 13,000 g for 5 min. To isolate proteins from the intestinal lumen, the entire gastrointestinal tract (from the duodenum to distal colon) was recovered from five wild-types C57BL/6 J mice. The intestines were flushed with ~50 ml cold PBS containing protease inhibitors (Millipore Sigma, 11836153001). The flushes and fecal pellets were homogenized by rotor and stator (TH Tissue Homogenizer; OMNI; TH01) and large particles were centrifuged at 100 g for 10 min at room temperature. The supernatants were carefully decanted and centrifuged further at 3000 g for 20 min at room temperature. The clarified supernatants were precipitated with 40% ammonium sulfate overnight at 4 °C. Precipitated protein was centrifuged at 3000 g for 30 min at 4 °C, then resuspended in cold 40% ammonium sulfate and centrifuged again. The pellets were resuspended in room temperature PBS and allowed to mix for 10 min. Protein concentrations were determined by Bradford assay (BioRad).

## Immunoblot

50 µg of fecal protein or 25 µg of cellular protein was loaded onto a 4–20% gradient SDS-PAGE and transferred to a PVDF membrane. Membranes were blocked in 5% nonfat dry milk in Tris-buffered saline (TBS) with 0.1% Tween-20 and then incubated overnight with the following primary antibodies: anti-C1Q (PA5-29586, Thermo Fisher) and anti-actin (13E5, Cell Signaling). REG3G was detected by incubating membranes with rabbit anti-REG3G antiserum (*Cash et al., 2006*). After washing, membranes were incubated with goat anti-rabbit IgG HRP and then visualized with a BioRad ChemiDoc Touch system.

## Enzyme-linked immunosorbent assay (ELISA)

Mouse C1q ELISA was performed as previously described (*Petry et al., 2001*). Briefly, microtiter plates were coated overnight with mouse IgG1 and were then blocked with 5% BSA in PBS. Serum samples were diluted 1:50 and plated for 1 hr at room temperature. After washing with 0.05% Tween-20 in PBS, bound C1q was incubated with a biotinylated anti-C1q antibody (JL1, Abcam). Biotinylated anti-C1q was detected with a streptavidin-HRP conjugate (Abcam). Optical density was measured using a wavelength of 492 nm. Plates were analyzed using a SpectraMax M5 microplate reader (Molecular Devices).

## Intestinal permeability assay

Intestinal permeability assays were performed by treating mice with fluorescein isothiocyanate dextran (FITC-dextran; 4000 Da) by oral gavage. The non-steroidal anti-inflammatory drug (NSAID) indomethacin was administered to mice as a positive control. For the experimental group, mice were treated with 190 µl 7% dimethyl sulfoxide (DMSO) in PBS by oral gavage. For the positive control group, mice were treated with 190 µl indomethacin (1.5 mg/ml in 7% DMSO in PBS) by oral gavage. After 1 hr, all mice were treated with 190 µl FITC-dextran (80 mg/ml in PBS) by oral gavage. Mice were sacrificed after 4 hr and sera were collected. Serum samples were centrifuged for 20 min at 4 °C at 800 g and supernatants were collected. Serum FITC-dextran levels were measured by a fluorescence microplate assay against a standard curve using a Spectramax plate reader (Molecular Devices).

## 16*S* rRNA gene quantification (absolute copy number)

Age and sex-matched mice were sacrificed and mesenteric lymph nodes were harvested and weighed. Total DNA was extracted using the Qiagen DNEasy kit. Microbial genomic DNA was quantified against a standard curve by qPCR analysis using universal 16*S* rRNA gene primers and the SsoAdvanced SYBR Green Supermix (BioRad). Total copy numbers of bacterial 16*S* RNA genes were normalized to tissue weight.

## Dextran sulfate sodium (DSS) treatment

Age and sex-matched mice were provided with 3% dextran sulfate sodium (weight/volume) in autoclaved drinking water for seven days. Animal weight and health were monitored in accordance with institutional IACUC guidelines. On day 7, animals were sacrificed and colon lengths were recorded. Terminal colon segments were fixed in Bouin's fixative for 24 hr followed by washes in 70% ethanol. Tissues were paraffin-embedded and sectioned by the UT Southwestern Histopathology Core facility. Tissue specimens were scored by a pathologist who was blinded as to the mouse genotypes. Disease severity was scored using five different parameters on a scale of 0–4: inflammation severity, edema severity, epithelial cell loss severity, hyperplasia, and fibrosis. Scores for each individual parameter were added together to represent the overall histology score.

## *Salmonella typhimurium* infection

To prepare bacteria for infection, *Salmonella enterica* serovar *typhimurium* (SL1344) was cultured in Luria-Bertani (LB) broth containing 50 µg/ml streptomycin in a shaking incubator at 37 °C (*Eichelberg and Galán, 1999*). The overnight culture was diluted the next day and grown to the mid-log phase (OD$_{600}$ = 0.3–0.5). *C1qa*$^{fl/fl}$ and *C1qa*$^{\Delta M\phi}$ littermates were inoculated intragastrically with $10^9$ CFU. All mice were sacrificed 24 hr post-infection and small intestinal tissues were harvested for analysis.

### *Citrobacter rodentium* infection

To prepare bacteria for infection, an overnight culture of *C. rodentium* (DBS100, ATCC) was grown in LB broth containing nalidixic acid (100 μg/ml) in a shaking incubator at 37 °C. The culture was diluted the next day and grown to the mid-log phase ($OD_{600}$ = 0.4–0.6). Bacteria were pelleted, washed, and resuspended in PBS. Sex-matched littermates were inoculated intragastrically with $5 \times 10^8$ CFU. Fecal pellets were collected at a fixed time every 48 hr, homogenized in sterile PBS, diluted, and plated on LB agar with nalidixic acid (100 μg/ml).

## Immunofluorescence analysis of mouse intestines

Mouse small intestines and colons were flushed with PBS and embedded with Optimal Cutting Temperature compound (OCT) (Thermo Fisher). Sections were fixed in ice-cold acetone, blocked with 1% BSA, 10% FBS, 1% Triton X-100 in PBS, and then incubated overnight at 4 °C with the following antibodies: mouse anti-C1q biotin (JL-1), rat anti-CD169 (3D6.112), and rabbit anti-TUBB3 (ab18207, Abcam). Slides were then washed with PBS containing 0.2% Tween-20 (PBS-T) and incubated with donkey anti-rabbit AlexaFluor 488, donkey anti-rat AlexaFluor 594, and Streptavidin-Cy5 (Thermo Fisher) for 1 hr at room temperature in the dark. Slides were then washed in PBS-T and mounted with DAPI-Fluoromount-G (Southern Biotech). Mounted slides were cured overnight at 4 °C until imaging.

For immunofluorescence analysis of longitudinal muscle-myenteric plexus wholemounts, intestines were prepared by first removing the adipose tissues and flushing the luminal contents. A 1 ml pipette tip was inserted into the intestinal lumen to fully extend the intestinal wall. The longitudinal muscle-myenteric plexus layer was then separated from the mucosa using cotton swabs as previously described (*Ahrends et al., 2022*; *Obata et al., 2020*). The longitudinal muscle-myenteric plexus layer was then stretched by pinning the tissues on a Sylgard-coated Petri dish (Fisher Scientific) containing cold PBS and fixed with 4% PFA overnight at 4 °C. The fixed tissues were rinsed five times with PBS at room temperature with shaking and then permeabilized and blocked with PBS containing 1% Triton X-100 and 10% normal donkey serum (NDS) for 1 hr at room temperature. The tissues were incubated with primary antibodies in the same solution overnight at 4 °C. The tissues were then washed with PBS containing 1% Triton X-100 and incubated with secondary antibodies in the blocking buffer for 2 hr at room temperature. Immunostained tissues were washed four times with PBS containing 1% Triton X-100. After a final wash with PBS, tissues were mounted on Superfrost Microscope Slides using VECTASHIELD (Vector Laboratories).

## RNAscope analysis

Fluorescence in situ hybridization on the longitudinal muscle-myenteric plexus was carried out using the Advanced Cell Diagnostics RNAscope Fluorescent Multiplex Kit according to the manufacturer's instructions with some modifications as described previously (*Obata et al., 2020*; *Obata et al., 2022*). After hybridization, tissues were counterstained for neuronal nuclei as previously described and mounted on Superfrost Microscope Slides (Fisher Scientific) using VECTASHIELD (Vector Laboratories).

## Image processing

Fluorescently labeled longitudinal muscle-myenteric plexus preparations were imaged by a spinning disk confocal microscope (Nikon) with a Hamamatsu Orca-Fusion sCMOS camera using the NIS-Elements Advanced Research software (Nikon). All image analyses were performed using the image-processing package Fiji and ImageJ. The number of HuC/D$^+$ neurons in the myenteric plexus was quantified using a semi-automated image analysis pipeline Gut Analysis Toolbox (*Sorensen et al., 2022*).

## RNA-seq analysis of colonic longitudinal muscle-myenteric plexus

The colonic longitudinal muscle-myenteric plexus was collected from five age-matched male $C1qa^{fl/fl}$ and $C1qa^{\Delta M\phi}$ mice by manual dissection using a 2 mm metal probe (Fisher Scientific). RNA was isolated using the RNeasy Mini kit according to the manufacturer's protocol (Qiagen). Quantity and quality of RNA samples were assessed on a Bioanalyzer 2100 (Agilent Technologies). RNA-seq libraries were prepared using the TruSeq RNA sample preparation kit (Illumina) according to the manufacturer's protocol. Libraries were validated on a Bioanalyzer 2100 (Agilent Technologies). Indexed libraries were sequenced on an Illumina NextSeq550 for single-end 75 bp length reads. CLC Genomics Workbench

7 was used for bioinformatics and statistical analysis of the sequencing data. The approach used by CLC Genomics Workbench is based on a method developed previously (*Mortazavi et al., 2008*). To identify differentially enriched biological pathways, all genes were ranked based on their $\log_2$ fold-change, and pathway enrichment was identified using the R packages 'clusterProfiler' and 'msigdbr.' For analysis of differentially expressed genes, gene counts were analyzed using DESeq-2, and differentially expressed genes were defined as having an adjusted p-value < 0.05. A Fisher's Exact Test was conducted to assess the overlap between differentially expressed genes in $C1qa^{\Delta M\phi}$ mice and the TashT mouse (*Bergeron et al., 2015*).

## Single-cell RNA sequencing (scRNAseq) analysis

Single-cell RNA sequencing was done in the Microbiome Research Laboratory at UT Southwestern Medical Center. Lamina propria cell suspensions were prepared as previously described (*Yu et al., 2013*; *Yu et al., 2014*) from the small intestines of three $C1qa^{fl/fl}$ and three $C1qa^{\Delta M\phi}$ littermates. Total small intestinal cells were pooled according to genotype and live CD45+CD11b+MHCI-I+F4/80hi macrophages were sorted using a FACSAria (BD Biosciences). 5000–10,000 macrophages from each genotype with a viability score of >70% were input into each library. A 10 X Genomics Chromium controller instrument was used for Gel Bead-in Emulsion (GEMs) preparation. Chromium Next GEM Single Cell 3' Kit v3.1 (PC-1000269), Chromium Next GEM Chip G Single Cell Kit (PC-1000127), and Dual Index Kit TT Set A Kit (PC-1000215) were used for single-cell library preparation. cDNA and final barcoded sequencing libraries were generated according to the manufacturer's specifications and their quality and concentration were assessed using a Bioanalyzer 2100 (Agilent Technologies) and qPCR, respectively. Single-cell libraries that passed the quality checks were sequenced on a NextSeq550 sequencer using a paired-end 75 bp High Output sequencing kit. About 20,000–30,000 sequencing reads were generated per single cell. Unique molecular identifier (UMI) counts for each cellular barcode were quantified and used to estimate the number of cells successfully captured and sequenced. The Cell Ranger Single-Cell Software suite (10 X Genomics) was used for demultiplexing, barcode processing, alignment, and initial clustering of the raw scRNAseq profiles.

The Seurat V3 R package was used to filter and analyze the Cell Ranger output (*Stuart et al., 2019*). Features that were in less than three cells and cells with less than 50 features were first filtered. To filter out dead or dying single cells, only cells that expressed more than 200 but less than 2500 features and cells in which mitochondrial transcripts accounted for less than five percent of all cell transcripts were used for further analysis. The single-cell data of these high-quality cells was then log-normalized and scaled. For further correction, the percentage of transcripts from mitochondria was regressed out. Dimension reduction was performed in Seurat and further differential gene expression was performed using *limma* (*Ritchie et al., 2015*). Pathway enrichment analysis was performed with Gene Set Enrichment Analysis (GSEA) via clusterProfiler (*Yu et al., 2012*). Visual representations of data were made using ggplot2 and Seurat R packages (*Love et al., 2015*).

## 16S rRNA gene sequencing and analysis

The hypervariable regions V3 and V4 of the bacterial 16S rRNA gene were prepared using the Illumina Nextera protocol (Part # 15044223 Rev. B). An amplicon of 460 bp was amplified using the 16S Forward Primer and 16S Reverse Primer as described in the manufacturer's protocol. Primer sequences are given in the Key Resources Table. The PCR product was purified using Agencourt AmpureXP beads (Beckman Coulter Genomics). Illumina adapter and barcode sequences were ligated to the amplicon to attach them to the MiSeqDx flow cell and for multiplexing. Quality and quantity of each sequencing library were assessed using Bioanalyzer (Agilent Technologies) and Picogreen (Thermo Fisher) measurements, respectively. Libraries were loaded onto a MiSeqDX flow cell and sequenced using the Paired End 300 (PE300) v3 kit. Raw fastq files were demultiplexed based on unique barcodes and assessed for quality. Samples with more than 50,000 quality control pass sequencing reads were used for downstream analysis. Taxonomic classification and operational taxonomic unit analysis were done using the CLC Microbial Genomics Module. Individual sample reads were annotated with the Greengene database and taxonomic features were assessed.

## Gastrointestinal motility assays

Motility assays were adapted from previous studies (*Luo et al., 2018*; *Maurer, 2016*; *Muller et al., 2014*). To determine transit time through the entire gastrointestinal tract, age-matched male mice were fasted overnight and water was removed 1 hr prior to the start of the experiment. Mice were then singly housed for 1 hr and then gavaged with 100 μl of Carmine Red (5% weight/volume; Sigma) in 1.5% methylcellulose. Fecal pellets were collected every 15 min and transit time was recorded when the dye was first observed in the feces.

For small intestinal motility measurements, age-matched male mice were fasted overnight and then gavaged with 100 μl of rhodamine B-dextran (5 mg/ml; Thermo Fisher) in 2% methylcellulose. After 90 min, mice were sacrificed and their stomachs, small intestines, ceca, and colons were collected. Small intestines were cut into eight segments of equal length and colons were cut into five segments of equal length. Segments were cut open lengthwise and vortexed in 1 ml PBS to release rhodamine B-dextran. Fluorescence was then measured on a SpectraMax M5 microplate reader (Molecular Devices). The geometric center of the dye was calculated as: GC = Σ (% of total fluorescent signal per segment × segment number). Relative fluorescence per segment was calculated as: (fluorescence signal in segment/total fluorescence recovered) × 100.

To measure colonic motility, age-matched male mice were fasted overnight and lightly anesthetized with isoflurane. A 2 mm glass bead was inserted 2 cm intrarectally using a 2 mm surgical probe. Mice were then returned to empty cages and the time to reappearance of the bead was recorded.

To account for potential circadian differences in gut motility, the time of day for the initiation of all experiments was held constant.

## Ex vivo peristaltic imaging

Ex vivo video imaging and analysis of colonic peristalsis were carried out as described previously (*Obata et al., 2020*) on age-matched male mice. Colons were dissected, flushed with sterile PBS, and pinned into an organ bath chamber (Tokai Hit, Japan) filled with Dulbecco's Modified Eagle Medium (DMEM). DMEM was oxygenated (95% $O_2$ and 5% $CO_2$), run through the chamber using a peristaltic pump (MINIPULS 3, Gilson), and kept at 37 °C. Colons were allowed to equilibrate to the organ chamber for 20 min before video recording. Time-lapse images of colonic peristalsis were captured with a camera (MOMENT, Teledyne photometrics) using PVCAM software (500 ms time-lapse delay) and recorded for 45 min.

For analysis of colonic migrating motor complexes (CMMC), videos consisting of 5400 sequential image frames were stitched together in Fiji and read into Igor Pro 9 (WaveMetrics) to generate spatiotemporal maps using a customized algorithm developed by the Pieter Vanden Berghe lab at the University of Leuven, Belgium (*Roosen et al., 2012*). The generated spatiotemporal maps were used to determine the frequency and period of CMMCs. Each CMMC on the spatiotemporal map was further projected onto the axes to obtain the distance traveled (millimeters) and the time for the CMMC to travel such distance (seconds), allowing us to calculate the velocity (millimeter/second) of CMMCs.

## Statistical analysis

Graphed data are presented as means ± standard error of the mean (SEM). Statistics were determined with GraphPad Prism software. Statistical analyses were performed using a two-tailed Student's *t*-test when comparing two groups, oneway ANOVA when comparing multiple groups, and Fisher's exact test to assess overlap between groups of differentially expressed genes. The statistical tests used are indicated in the figure legends. *p<0.05; **p<0.01; ***p<0.001; ****p<0.0001; and ns, not significant (p>0.05).

## Acknowledgements

We thank Shai Bel for assistance with immunofluorescence imaging experiments, the UT Southwestern Genomics Core for assistance with RNA sequencing experiments, the UT Southwestern Flow Cytometry Core for assistance with flow cytometry experiments, Bret Evers (UT Southwestern Histo Pathology Core) for pathology scoring, and the Quantitative Light Microscopy Core (QLMC), a Shared Resource of the Harold C Simmons Cancer Center. The QLMC is supported in part by the National

Cancer Institute Cancer Center Support Grant P30 CA142543-01 and NIH 1S10OD028630-01. *Citrobacter rodentium* strain DBS100 was a gift from Vanessa Sperandio (UT Southwestern). The laboratory of Pieter Vanden Berghe (University of Leuven, Belgium) provided the algorithm used to generate spatiotemporal maps of colonic migrating motor complexes. This work was supported by NIH grants R01 DK070855 (LVH), Welch Foundation Grant I-1874 (LVH), the Walter M and Helen D Bader Center for Research on Arthritis and Autoimmune Diseases (LVH), and the Howard Hughes Medical Institute (LVH). MP was supported by NIH T32 AI005284. AAC was supported by NIH T32 AI005284 and NIH F32 DK132913. EK was supported by NIH F31 DK126391. YO is the Nancy Cain Marcus and Jeffrey A Marcus Scholar in Medical Research, in Honor of Dr. Bill S Vowell.

## Additional information

### Funding

| Funder | Grant reference number | Author |
| --- | --- | --- |
| National Institutes of Health | R01 DK070855 | Lora V Hooper |
| Welch Foundation | I-1874 | Lora V Hooper |
| Howard Hughes Medical Institute | | Lora V Hooper |
| National Institutes of Health | T32 AI005284 | Mihir Pendse |
| National Institutes of Health | F32 DK132913 | Alexander A Crofts |
| National Institutes of Health | F31 DK126391 | Eugene Koo |

The funders had no role in study design, data collection and interpretation, or the decision to submit the work for publication.

### Author contributions

Mihir Pendse, Conceptualization, Data curation, Formal analysis, Supervision, Investigation, Methodology, Writing – original draft, Writing – review and editing; Haley De Selle, Nguyen Vo, Data curation, Formal analysis, Investigation, Methodology; Gabriella Quinn, Alexander A Crofts, Data curation, Formal analysis; Chaitanya Dende, Daniel C Propheter, Investigation, Writing – review and editing; Yun Li, Cristine N Salinas, Tarun Srinivasan, Brian Hassell, Kelly A Ruhn, Investigation; Eugene Koo, Investigation, Methodology; Prithvi Raj, Data curation, Formal analysis, Investigation; Yuuki Obata, Investigation, Methodology, Writing – original draft, Writing – review and editing; Lora V Hooper, Conceptualization, Supervision, Funding acquisition, Writing – original draft, Project administration, Writing – review and editing

### Author ORCIDs

Mihir Pendse  http://orcid.org/0000-0002-7810-6791
Alexander A Crofts  http://orcid.org/0000-0003-0811-9199
Yuuki Obata  http://orcid.org/0000-0001-5461-3521
Lora V Hooper  http://orcid.org/0000-0002-2759-4641

### Ethics

This study was performed in strict accordance with the recommendations in the Guide for the Care and Use of Laboratory Animals of the National Institutes of Health. All of the animals were handled according to approved institutional animal care and use committee (IACUC) protocols (protocol #2015-101212) of the University of Texas Southwestern Medical Center.

### Decision letter and Author response

Decision letter https://doi.org/10.7554/eLife.78558.sa1
Author response https://doi.org/10.7554/eLife.78558.sa2

## Additional files

### Supplementary files
• MDAR checklist

### Data availability

16S rRNA gene sequencing data (Figure 3D) and RNA sequencing data (Figure 6A and B; Figure 1—figure supplement 1; Figure 6—figure supplement 1) are available from the Sequence Read Archive under BioProject ID PRJNA793870. All mouse strains used are available commercially.

The following dataset was generated:

| Author(s) | Year | Dataset title | Dataset URL | Database and Identifier |
|---|---|---|---|---|
| Pendse M, Raj P, Hooper LV | 2022 | Macrophages control gastrointestinal motility through complement component 1q | https://www.ncbi.nlm.nih.gov/bioproject/PRJNA793870/ | NCBI BioProject, PRJNA793870 |

The following previously published dataset was used:

| Author(s) | Year | Dataset title | Dataset URL | Database and Identifier |
|---|---|---|---|---|
| Gattu S, Bang Y, Chara A, Harris T, Kuang Z, Ruhn K, Sockanathan S, Hooper LV | 2019 | Epithelial retinoic acid receptor beta regulates serum amyloid A expression and vitamin A-dependent intestinal immunity | https://www.ncbi.nlm.nih.gov/geo/query/acc.cgi?acc=GSE122471 | NCBI Gene Expression Omnibus, GSE122471 |

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
