## [Editor Report]

This study provides a fundamental finding that complement C1q produced by enteric macrophages shapes neuronal function and gut motility. The authors present convincing data showing that while macrophage-derived C1q is not necessary for defenses against enteric pathogens, it plays an important role in regulating neuronal gene expression and intestinal transit. These findings will be of interest to gastroenterologists, neuroscientists and immunologists in revealing a novel neuroimmune axis in gut homeostasis.

---

## [Decision Letter]

**Decision letter after peer review:**

Thank you for submitting your article "Macrophages regulate gastrointestinal motility through complement component 1q" for consideration by *eLife*. Your article has been reviewed by 3 peer reviewers, one of whom is a member of our Board of Reviewing Editors, and the evaluation has been overseen by Wendy Garrett as the Senior Editor. The following individual involved in review of your submission has agreed to reveal their identity: Meenakshi Rao (Reviewer #3).

The reviewers have discussed their reviews with one another, and the Reviewing Editor and I have drafted this to help you prepare a revised submission.

Essential revisions:

1) An important question is mechanistically how C1q affects gut motility. One question is whether the ENS state changes. All 3 reviewers find it important to better characterize how enteric neurons are affected in the LysM-Cre C1q KO mice, which includes immunofluorescence analysis of neuronal subset numbers and health (Reviewer #1 point 1, Reviewer #2 point 3, Reviewer #3 point 4).

2) Reviewers also ask for stronger data to support the observation that C1q is mainly expressed by ENS-associated macrophages, which is a key point raised by this study. Better qPCR analysis of flow sorted macrophages is recommended by Reviewer #2 (point 1) and high resolution imaging analysis is suggested by Reviewer #3 (Reviewer #3, point 3). Reviewers #2-3 also ask how these C1q+ macrophages are localized in relation to myenteric plexus vs. submucosal plexus neurons. Whole mount analysis with confocal resolution and higher magnification is recommended.

3) The mechanism by which C1q could affect neurons is unclear. Reviewer #1 asked whether macrophage state is affected in the conditional KO mice. Reviewer #2 asked about potential expression of C1q sensors in the RNAseq datasets. Reviewer #3 also asked about secretomotor reflex and fluid flux, a major function of submucosal neurons.

4) Both reviewers #2-3 ask about specific variables that are not analyzed in the DSS colitis, *Salmonella* and citrobacter models.

5) Reviewer #3 brings up better characterization of LysM-Cre activity in the gut, and also other key variables such as whether there is sex-dependency of the effects.

We hope you find this synopsis of key areas to address helpful in your resubmission process.

*Reviewer #1 (Recommendations for the authors):*

1) Does loss of macrophage C1q alter enteric neuron development or numbers? In Figure 4E, RNAseq of longitudinal muscle-myenteric plexus shows changes at the transcriptional level for neuronal genes. Can the authors perform quantification of overall numbers of neurons in enteric ganglia, quantify major subsets (e.g. nitrergic or cholinergic) to see if the balance of proportions have changed in these mice?

2) While the hypothesis is that C1q likely acts on neurons, it is also possible that C1q is functioning in an autocrine manner to act on the macrophages. Have the authors characterized intestinal macrophages isolated from C1qa floxed vs. C1qa∆LysM mice to see if there are changes in gene expression?

*Reviewer #2 (Recommendations for the authors):*

1. There is no strong data supporting the notion that C1q is mostly expressed by ENS-associated muscularis macrophages (Figure 4). Tissue cross-sections are nice but they do not clearly reflect the distribution of macrophages in the GI tissue. For example, a subset of CD169+ macrophages is also present in the mucosa (PMID: 26193821) although it was never tested whether CD169+ mucosal macrophages are associated with nerve fibers. According to the Immgen dataset (Immgen.org, also used in PMID: 25036630), a comparison of C1q gene expression by mucosal vs muscularis (serosal) macrophages shows equal levels of C1q expression. Therefore, a more direct demonstration of C1q expression by muscularis macrophages as compared to mucosal macrophages would be needed. For this, qPCR to measure C1q expression should be done in macrophages isolated from the mucosa vs separated muscle layer (with the myenteric plexus) along with flow cytometry for C1q.

2. The major weakness of the study is that there is no direct evidence that C1q acts on the cells that compose the ENS (e.g., on neurons, enteric glia, or even macrophages in an autocrine manner), particularly if it is confirmed that both mucosal and muscularis macrophages express C1q. Is it possible to provide a more comprehensive analysis of the expression of C1q sensor(s) by neurons or other ENS cell types, at least in a cell culture system, to confirm this connection and to test the functional outcome(s) of their activation? The RNAseq dataset from Cre/flox vs control mice (Figure 5E, F) could provide important clues.

3. Immunofluorescence of whole tissue mounts of the muscle layer (instead of cross-sections shown in Figure 4) would provide a more comprehensive analysis of the ENS state and macrophage-ENS interactions in Lyz2-Cre:C1q-fl/fl mice as compared to controls. It may reveal some structural abnormalities of the ENS explaining the phenotype.

4. Figure 3C, experiment with *Salmonella* infection: 24 hr time point might be too early. For example, detectable changes in macrophage gene expression appear on Days 3-5 post-infection.

5. Figure 3H: What about expression of the first line of defense inflammatory genes such IL1a and IL1b, TNF, IL12a and IL12b, SAA etc highly expressed by gut macrophages?

6. Quantifying bacterial CFU in the mesenteric lymph nodes at steady state is a sensitive readout for assessing the integrity of mucosal barrier (could be included to support conclusions of Figure 3). The same readout can be applied to Citrobacter infection.

*Reviewer #3 (Recommendations for the authors):*

– Characterization of Lyz2-Cre activity at the tissue level in a Cre-dependent reporter with small intestine and colon analyzed in adult mice

– More thorough reporting of phenotypes in DSS colitis and Citrobacter infection models

– Improved imaging analysis of C1q expression with confocal level resolution and higher magnification, ideally of whole mounts rather than tissue sections in which neuronal and macrophage cell soma can be more clearly visualized. C1q immunoblot of LMMP preparations would also help clarify to what extent C1q is or is not present in muscularis externa.

– Analysis of fluid content in fecal pellets and measures of secretomotor reflex function in KO mice

---

## [Author Response]

Reviewer #1 (Recommendations for the authors):1) Does loss of macrophage C1q alter enteric neuron development or numbers? In Figure 4E, RNAseq of longitudinal muscle-myenteric plexus shows changes at the transcriptional level for neuronal genes. Can the authors perform quantification of overall numbers of neurons in enteric ganglia, quantify major subsets (e.g. nitrergic or cholinergic) to see if the balance of proportions have changed in these mice?

We agree that assessing the architecture of the enteric nervous system is essential for understanding C1q’s role in regulating gastrointestinal motility. We therefore determined the total number of enteric neurons per unit area. We found no statistically significant differences in neuron numbers in *C1qa*^ΔMφ^ mice when compared to *C1qa*^fl/fl^ littermates. As suggested by the reviewer, we also assessed whether the *C1qa*^ΔMφ^ mice have altered numbers of inhibitory (nitrergic) and excitatory (cholinergic) neuronal subsets (Figure 5). Here we used both RNAscope and qPCR analysis to measure expression of *Nos1*, which marks nitrergic neurons, and *Chat1*, which marks cholinergic neurons. Our findings indicate that numbers of inhibitory and excitatory neurons are essentially unchanged in the absence of macrophage C1q. We also did not detect changes in the enteric glial network marked by S100B. We present these results in a new Figure 5 and discuss them in the Results section of our manuscript.

2) While the hypothesis is that C1q likely acts on neurons, it is also possible that C1q is functioning in an autocrine manner to act on the macrophages. Have the authors characterized intestinal macrophages isolated from C1qa floxed vs. C1qa∆LysM mice to see if there are changes in gene expression?

This is an excellent question that intrigued us as well. To address it we conducted a single cell RNAseq experiment comparing small intestinal macrophages from *C1q^fl/fl^* and *C1qa*^ΔMφ^ mice (Figure 6 —figure supplement 3). Our analysis of the differentially expressed genes across all macrophage clusters indicated lowered representation of several transcripts that are linked to control of macrophage differentiation or functional state. Furthermore, in macrophages from *C1qa*^fl/fl^ mice, we observed expression of eight “microglia-specific genes” whose expression was lowered or lost in macrophages from *C1qa*^ΔMφ^ mice. Thus, it is possible that altered intestinal motility could arise in part from cell intrinsic functional alterations in C1q-deficient intestinal macrophages. However, it is not yet clear whether this arises from a C1q autocrine signaling loop or whether C1q imprints a neuronal function that feeds back to regulate macrophage gene expression and function. This is the subject of ongoing investigation in our lab.

Reviewer #2 (Recommendations for the authors):1. There is no strong data supporting the notion that C1q is mostly expressed by ENS-associated muscularis macrophages (Figure 4). Tissue cross-sections are nice but they do not clearly reflect the distribution of macrophages in the GI tissue. For example, a subset of CD169+ macrophages is also present in the mucosa (PMID: 26193821) although it was never tested whether CD169+ mucosal macrophages are associated with nerve fibers. According to the Immgen dataset (Immgen.org, also used in PMID: 25036630), a comparison of C1q gene expression by mucosal vs muscularis (serosal) macrophages shows equal levels of C1q expression. Therefore, a more direct demonstration of C1q expression by muscularis macrophages as compared to mucosal macrophages would be needed. For this, qPCR to measure C1q expression should be done in macrophages isolated from the mucosa vs separated muscle layer (with the myenteric plexus) along with flow cytometry for C1q.

Our original manuscript claimed that C1q-expressing macrophages were mostly located near enteric neurons in the submucosal plexus. However, as the Reviewer points out, this conclusion was based solely on our immunofluorescence analysis of tissue cross-sections. Reviewer 3 raised a similar concern, further pointing out that while the myenteric plexus is important for gut motility there is limited evidence for the involvement of the submucosal plexus.

As suggested, we further characterized C1q^+^ macrophage localization by flow cytometry analysis of macrophages isolated from the mucosa (encompassing both the lamina propria and submucosa) and the muscularis. Here we used the protocol provided by Ahrends et al. (*Star Protocols* 3, 101157) for separating the muscularis from the submucosal tissues. We found similar levels of C1q expression in macrophages from both tissues (Figure 4A in the revised manuscript). These findings agree with the IMMGEN analysis cited by the reviewer.

Although the mucosal macrophage fraction yielded by the Ahrends et al. protocol encompasses both lamina propria and submucosal macrophages, our immunofluorescence analysis suggests that the mucosal C1q-expressing macrophages are mostly from the submucosa, near the submucosal plexus (Figure 4B and C). This observation is consistent with the immunofluorescence studies of CD169^+^ macrophages shown in Asano et al., which suggest that most CD169^+^ macrophages are located in or near the submucosal region, with fewer near the villus tips (Figure 1e, *Nat. Commun.* 6, 7802).

Most importantly, our flow cytometry analysis indicates that the muscularis/myenteric plexus harbors C1q-expressing macrophages. To further characterize C1q expression in the muscularis, we performed RNAscope analysis by confocal microscopy of the myenteric plexus from mouse small intestine and colon (Figure 4D). The results show numerous C1q-expressing macrophages positioned close to myenteric plexus neurons and are thus consistent with our flow cytometry analysis. We note that although the majority of C1q immunofluorescence in our tissue cross-sections was observed near the submucosal plexus, we did observe some C1q expression in the muscularis by immunofluorescence (Figure 4B and C). Importantly, these new findings support the altered gut motility observed in the *C1qa*^∆Mφ^ mice. We have rewritten the Results section to take these new findings into account.

2. The major weakness of the study is that there is no direct evidence that C1q acts on the cells that compose the ENS (e.g., on neurons, enteric glia, or even macrophages in an autocrine manner), particularly if it is confirmed that both mucosal and muscularis macrophages express C1q. Is it possible to provide a more comprehensive analysis of the expression of C1q sensor(s) by neurons or other ENS cell types, at least in a cell culture system, to confirm this connection and to test the functional outcome(s) of their activation? The RNAseq dataset from Cre/flox vs control mice (Figure 5E, F) could provide important clues.

As suggested by the reviewer, we have further explored the expression of C1q receptors on enteric neurons. The prior study of Benavente et al. (2020) identified five possible transmembrane receptors for C1q in human neural stem cells. We therefore searched for the mouse homologs of the genes encoding these receptors in the dataset published in Obata et al. (2020), which compared the transcriptomes of mouse myenteric plexus neurons with those of non-neuronal cells and detected expression of four of the five receptors. We now show these data in Figure 6 —figure supplement 2. The gene encoding BAI1 (*Adgrb1*) was unique among these four genes by being more highly expressed in neurons as compared to non-neurons in both the small intestine and the colon (Figure 6 —figure supplement 2A). We further validated the expression of *Adgrb1* in enteric neurons of the myenteric plexus using RNAscope (Figure 6 —figure supplement 2B). These data suggest that C1q could interact directly with enteric neurons of the myenteric plexus that express BAI1.

We are currently working to test the functional outcomes of C1q-BAI1 interactions, with the aim of uncovering the molecular mechanisms by which C1q impacts enteric neurons and regulates gut motility. However, this involves generating new mouse models which will require many months, and thus we plan to include these findings in a follow-up manuscript.

3. Immunofluorescence of whole tissue mounts of the muscle layer (instead of cross-sections shown in Figure 4) would provide a more comprehensive analysis of the ENS state and macrophage-ENS interactions in Lyz2-Cre:C1q-fl/fl mice as compared to controls. It may reveal some structural abnormalities of the ENS explaining the phenotype.

We agree with both Reviewer 1 and Reviewer 2 regarding the need to carefully analyze the ENS state in *C1qa*^ΔMφ^ mice. These data are now provided in a new Figure 5. To summarize, we find no obvious anatomical change in the ENS as measured by counting total neurons and analysis of numbers of inhibitory and excitatory neuronal subsets. However, our RNAseq data on LMMP (Figure 6A and B) suggest that there are changes in neuronal function in the *C1qa*^ΔMφ^ mice. This idea is supported by increased neurogenic activity of peristalsis (Figure 6H and I) and the expression of the C1q receptor BAI1 on enteric neurons (Figure 6 —figure supplement 3).

4. Figure 3C, experiment with Salmonella infection: 24 hr time point might be too early. For example, detectable changes in macrophage gene expression appear on Days 3-5 post-infection.

We assayed C1q at 24-hours after oral *Salmonella* infection because this is the timepoint at which we typically see maximal expression of most intestinal antimicrobial proteins. We agree that longer infection times might reveal *Salmonella*-induced C1q expression. However, when we infected mice with *Salmonella* for >3 days we encountered issues with pathology and mortality that made our results difficult to interpret. For this reason, we have added phrasing to our description of the Results indicating this limitation of our interpretation:

“Although we cannot rule out induction of C1q by longer-term pathogenic infections, these data indicate that C1q is not induced by the gut microbiota or by a 24-hour infection with S. Typhimurium, in contrast to other intestinal antibacterial proteins.”

5. Figure 3H: What about expression of the first line of defense inflammatory genes such IL1a and IL1b, TNF, IL12a and IL12b, SAA etc highly expressed by gut macrophages?

We have added data to Figure 3H showing that a broader array of genes encoding secreted immune effectors, including innate cytokines and SAA proteins, are unchanged in *C1qa*^ΔMφ^ mice compared to *C1qa*^fl/fl^ controls.

6. Quantifying bacterial CFU in the mesenteric lymph nodes at steady state is a sensitive readout for assessing the integrity of mucosal barrier (could be included to support conclusions of Figure 3). The same readout can be applied to Citrobacter infection.

This is an excellent suggestion. To assay for bacterial translocation we analyzed bacterial 16*S* gene copy number in the mesenteric lymph nodes of *C1qa*^ΔMφ^ mice compared to *C1qa*^fl/fl^ controls. We found no statistically significant difference between the two mouse genotypes, providing further evidence that intestinal barrier function is largely intact in *C1qa*^ΔMφ^ mice.

Reviewer #3 (Recommendations for the authors):– Characterization of Lyz2-Cre activity at the tissue level in a Cre-dependent reporter with small intestine and colon analyzed in adult mice

As indicated by the reviewer, Cre expression driven by the *Lyz2* promoter is restricted to macrophages and some myeloid cells in the circulation (Clausen et al., 1999). To better understand intestinal *Lyz2* expression at a cellular level, we analyzed *Lyz2* transcripts from a published single cell RNAseq analysis of intestinal cells (Xu et al., 2019). These data show that intestinal *Lyz2* is also predominantly expressed in gut macrophages with limited expression in dendritic cells and neutrophils.

Additionally, our study shows that intestinal C1q expression is restricted to macrophages (CD11b^+^MHCII^+^F4/80^hi^) and is absent from other gut myeloid cell lineages (Figure 1E-H). This conclusion is supported by our finding that macrophage depletion via anti-CSF1R treatment also depletes most intestinal C1q (Figure 2A-C). Importantly, we found that the *C1qa*^ΔMφ^ mice retain C1q expression in the central nervous system (Figure 2 —figure supplement 1). Thus, the *C1qa*^ΔMφ^ mice allow us to assess the function of macrophage C1q in the gut and uncouple the functions of macrophage C1q from those of C1q in the central nervous system.

– More thorough reporting of phenotypes in DSS colitis and Citrobacter infection models

As outlined in our response to Point 2 above, we have provided additional histological analysis of the phenotypes in the DSS colitis and *Citrobacter* infection models.

– Improved imaging analysis of C1q expression with confocal level resolution and higher magnification, ideally of whole mounts rather than tissue sections in which neuronal and macrophage cell soma can be more clearly visualized. C1q immunoblot of LMMP preparations would also help clarify to what extent C1q is or is not present in muscularis externa.

We have addressed this in our response to Point 3 above and provide new data showing C1q expression in macrophages localized to the LMMP.

– Analysis of fluid content in fecal pellets and measures of secretomotor reflex function in KO mice

Our revised manuscript provides new data showing C1q expression by muscularis macrophages in the myenteric plexus. We analyzed muscularis macrophages by flow cytometry and found that they express C1q (Figure 4 —figure supplement 1). These findings are further supported by RNAscope analysis of C1q expression in wholemounts of LMMP from small intestine and colon (Figure 4D and E). These results are thus consistent with the increased CMMC activity and accelerated gut motility in the *C1qa*^ΔMφ^ mice. As suggested by the reviewer, our finding of C1q^+^ macrophages in the submucosal plexus indicates that C1q may also have a role controlling the function of submucosal plexus neurons. We are further exploring this idea through extensive additional experimentation. Given the expanded scope of these studies, we are planning to include them in a follow-up manuscript.